# Improving the thermocline calculation over the global ocean

Emmanuel Romero[1], Leonardo Tenorio-Fernandez[2], Esther Portela[3,4,5], Jorge Montes-Aréchiga[6], and Laura Sánchez-Velasco[1]

[1]Instituto Politécnico Nacional-Centro Interdisciplinario de Ciencias Marinas (IPN-CICIMAR), Departamento de Oceanología, Av. IPN s/n, La Paz, B.C.S., 23096, México
[2]CONACyT-Instituto Politécnico Nacional-Centro Interdisciplinario de Ciencias Marinas (IPN-CICIMAR), Av. IPN s/n, La Paz, B.C.S, 23096, México
[3]School of Environmental Sciences, University of East Anglia, Norwich, United Kingdom
[4]Univ. Brest, Laboratoire d'Océanographie Physique et Spatiale, CNRS, IRD, Ifremer, Plouzané, France
[5]Institute for Marine and Antarctic Studies, University of Tasmania, Hobart 7001, Australia
[6]Universidad de Guadalajara, Departamento de Física, Gral. Marcelino García Barragán 1421, Olímpica, 44430 Guadalajara, Jal, México

**Correspondence:** Leonardo Tenorio-Fernandez (ltenoriof@ipn.mx)

**Abstract.** According to the typical thermal structure of the ocean, the water column can be divided into three layers: the mixed layer, the thermocline and the deep layer. In this study, we provide a new methodology, based on a function adjustment on the temperature profile, to locate the minimum and maximum depths of the strongest thermocline. We first validated our methodology by comparing the mixed layer depth obtained with the method proposed here with three other methods from previous studies. Since we found a very good agreement between the four methods we used the function adjustment to compute the monthly climatologies of the maximum thermocline depth, the thermocline thickness and strength, in the global ocean. We also provide an assessment of the regions of the ocean where our adjustment is valid, i.e. where the thermal structure of the ocean follows the three-layer structure. However, there are ocean regions where the water column cannot be separated into three layers due to the dynamic processes that alter it. This assessment highlights the limitations of the existing methods to accurately determine the mixed layer depth and the thermocline depth in oceanic regions that are particularly turbulent as the Southern Ocean and the northern North Atlantic, among others. The method proposed here has shown to be robust and easy to apply.

## 1 Introduction

In most of the ocean, a typical vertical temperature profile shows maximum temperature at the surface, due to solar radiation, and can be divided into three main layers according to the thermal structure of the ocean: (i) the mixed layer, where the turbulence generated by atmospheric processes homogenizes the temperature and distributes heat throughout the layer; (ii) the thermocline, the layer with the strongest stratification, that separates the upper mixed layer from the deep layer of the ocean; and (iii) the deep layer, where the temperature is practically invariant over time and relatively constant from the lower thermocline to the seafloor. This three-layer structure is similar to the fundamental vertical density structure of the world ocean (Sallée et al., 2021), where the central layer is the pycnocline.

In most of the ocean, the temperature exerts the main control on the density of the water column. Exceptions to this are mainly found in polar regions, where temperature is very low and the seawater density is mostly determined by salinity (de Boyer Montégut et al., 2004; Yamaguchi and Suga, 2019; Sallée et al., 2021), and in the so-called Barrier Layer (BL) regions (Lorbacher et al., 2006). The latter, are regions where the mixed layer depth (MLD) is determined by a halocline. In these regions, the MLD based on temperature (the isothermal layer) is deeper than MLD based on density profiles (the isopycnal layer). In the opposite case, when the isothermal layer is shallower than the MLD, the vertical compensation between salinity and temperature causes compensated layers (CL) located below the MLD (de Boyer Montégut et al., 2004).

The thermocline is the ocean layer where the temperature changes sharply with depth as compared to the upper and lower layers (Fiedler, 2010). Consequently, the thermocline depth is often defined as the depth of the maximum vertical temperature gradient. The characteristics and depth of this layer vary spatially. In low latitudes, due to relatively higher temperatures in the upper water column, the stratification is high and permanent thermoclines are relatively strong and thin. In contrast, at high latitudes, where there is generally little difference between the surface and deep layer temperature, the thermocline is generally weaker and deeper (Webb, 2021). The strengthening of the upper thermocline in mid-latitudes during summer, when net heat flux at the surface is positive and wind mixing is low, is known as the seasonal thermocline (Sprintall and Cronin, 2001). Due to cooling, wind-driven mixing, and a well-stratified thermocline, the mixed layer is deeper in winter (Sprintall and Cronin, 2001). In tropical and polar regions the seasonal changes are weak.

Other classifications of the thermocline have been proposed from a machine learning approach. For instance Jiang et al. (2017) classify the thermocline due to its form as positive, inverse and mixed thermoclines as well as multi-thermoclines. The forms that originate this classification could be related to the temperature inversions that occur at the base of the BL and in the polar regions (de Boyer Montégut et al., 2004; Dong et al., 2008) and by the double-diffusive staircase stratification events (Timmermans et al., 2008; Toole et al., 2011).

The MLD (which is also the top of the thermocline) as well as the maximum thermocline depth (MTD), and thermocline strength, all play a key role in determining the vertical distribution of many physical and ecological parameters (Fiedler, 2010). The thermocline is a physical gradient that plays a key role in climate variability and ocean-atmosphere interactions (Chu and Fan, 2019). The thermocline strength affects buoyancy, heat budgets, circulation, and exchange of properties. Its depth is associated with the habitat and abundance of zooplankton organisms (Southward and Barrett, 1983; Ruvalcaba-Aroche et al., 2022) and is also an ecological boundary for the pelagic organisms. Strong temperature changes can set habitat distributions and the thermocline often corresponds to gradients in nutrients (nutricline), oxygen (oxycline), or other limiting factors. The thermocline thickness also affects the intensity of the primary production. Particularly at the poles, where the thermocline is weaker, enhanced mixing distributes nutrients throughout the water column. In contrast, in equatorial and tropical regions, the strong thermocline prevents nutrient-rich water from the deep layer from reaching the surface (Webb, 2021). Observations of tracer concentrations and model simulations suggest a connection between the equatorial thermocline and mid-latitude ventilation regions (Harper, 2000).

The strength of the ocean stratification has strong implications for the ventilation of the interior ocean and the injection of traces as oxygen and carbon (Sallée et al., 2012; Portela et al., 2020). Therefore, the knowledge and monitoring of the depth

and strength of the thermocline is particularly relevant in the context of ocean warming and its effects in the pelagic ecosystem. Recent global studies have found an overall increase of the global ocean stratification (Li et al., 2020) as well as important regional variability in the global pycnocline trend over the past decades (Sallée et al., 2021). For instance, the thickness of the equatorial and tropical thermocline is enhanced under ocean warming, because the surface layer warms more and faster than the lower layers (Yang and Wang, 2009).

Previous regional studies have identified a shallowing and strengthening thermocline in the western Pacific Ocean (Vecchi and Soden, 2007) and in the equatorial Pacific (Zelle et al., 2004). Additionally, modeling studies have suggested that important changes in the Pacific Ocean, such as rising sea levels and temperatures, affect the structure of the thermocline from the subtropics to the tropics (Landerer et al., 2007; Overland and Wang, 2007). These changes in thermocline in the western tropical Pacific are considered to influence the properties of ENSO (Luo et al., 2009), which has strong climatic and socio-economical consequences at basin scale, because the organisms change their distribution and abundance.

Different methodologies have been proposed to locate the maximum depth of the mixed layer (e.g., de Boyer Montégut et al., 2004; Lorbacher et al., 2006; de Boyer Montégut et al., 2007; Holte and Talley, 2009) and the strength and trends of the ocean stratification (Yamaguchi and Suga, 2019; Li et al., 2020; Sallée et al., 2021). However, little efforts have been dedicated to identify and map the MTD on a global scale. This depth can be delimited empirically by locating the rapid temperature change in the profile, other studies are based on, for instance, calculating the thermocline gradient using the exponential leap-forward gradient method (Chu and Fan, 2017), the maximum curvature point method (Jiang et al., 2016), or using a matrix to calculate the temperature gradient strength of each point and filtering those points that meet the thermocline standard ($> 0.2\ °\mathrm{C}\ \mathrm{m}^{-1}$) (Jiang et al., 2017). Additionally, Fiedler (2010) has compared different methods to estimate the mixed layer depth, thermocline depth, and thermocline strength. In his study, the method that gave the best results was the Variable Representative Isotherm (VRI). This method locates the thermocline from the base of the mixed layer to the depth at which temperature has dropped halfway toward the deep-water temperature at 400 m (Fiedler, 2010). Despite few studies applying the above methods to particular regions of the ocean, to the best of our knowledge, there are no studies addressing the MTD on a global scale. The methodologies mentioned above to locate the MLD use data from the profilers of the Argo program. Argo is an international program that measures the ocean water column using a fleet of autonomous profilers, which move along ocean currents and measure the water column by making profiles from a depth of two kilometers to the surface (Argo, 2022a).

This paper proposes a simple and efficient methodology to locate the minimum and maximum depth of the thermocline and its thickness, making an adjustment of the sigmoid function to the temperature profiles. Locating these depths helps to conduct research on thermocline-related ocean warming and through the proposed methodology, it will be possible to conduct local and global studies on changes in ocean thermal structure through time and space.

In this study, we first describe the proposed method to calculate the MLD and MTD, then we compare the results with other methods found in the literature, finally we calculate the thickness and strength of the thermocline, to obtain the climatologies of the mixed layer depth, the maximum thermocline depth, the thermocline thickness, and the thermocline strength index.

## 2 Data collection

For all the diagnostics carried out in this study we used the Argo dataset. We downloaded the snapshot of January 2022 (Argo, 2022b) and we used the profiles already evaluated by the delayed mode quality control (DMQC) from January 1998 to December 2021 (more than two million), that have been classified as good or probably good data.

We selected pressure, temperature and salinity profiles from the core Argo floats, which typically sample down to 2000 m. We then transformed the in-situ temperature and practical salinity into conservative temperature ($\Theta$), and absolute salinity ($S_A$), using the definition of the Thermodynamic Equation of SeaWater 2010 (TEOS-10) (IOC et al., 2010).

## 3 Methodology

Previous studies (e.g., de Boyer Montégut et al., 2004; Holte and Talley, 2009) have proposed different methodologies to calculate the MLD on a global scale. Despite the existence of methods to calculate MTD (such as those compared in the revision study of Fiedler (2010)), these have been evaluated with a limited amount of data, and in relatively small tropical and subtropical areas, therefore excluding profiles of high latitudes. Here we propose to use a new method based on the sigmoid function adjustment in the temperature profile to localize the MTD. Our method takes advantage of the characteristics that this function shares with the typical temperature profiles in most of the ocean: a straight line that represents the homogeneity of the MLD, a diagonal that represents the rapid increase or decrease in temperature with depth (changing the sign of the function) in the thermocline, and a straight line that represents the little variability of the temperature of the deep ocean (Fig. 1).

To locate the MTD, we computed the vertical maximum of the contribution of temperature to the Brunt-Vaisala frequency squared (i.e. maximum of $N_T^2$) to locate the most stratified point from the temperature profile. We assume that this point is within the thermocline, as the most stratified point of the water column given by $N^2$ is inside the pycnocline (IOC et al., 2010). $N_T^2$ is given by Eq. (1), where $g$ is the gravitational acceleration, $\rho$ is the density, $\alpha^\Theta$ is the coefficient of thermal expansion, $\Delta\Theta$ is the difference between conservative temperatures of vertically adjacent seawater parcels separated in pressure by $\Delta P$ (IOC et al., 2010).

$$N_T^2 = g^2 \rho \frac{-\alpha^\Theta \Delta\Theta}{\Delta P} \tag{1}$$

Schematically, most of the temperature profiles in all latitudes have a shape similar to the sigmoid function (s-shape), for this study we used the logistic function shown in Eq. (2), where $a$ is the steepness of the curve and $b$ is the value of the midpoint of the function also known as the inflection point.

$$f(x) = \frac{1}{1 + e^{-a(x-b)}} \tag{2}$$

To perform the function adjustment, we first locate the greatest absolute value of $N_T^2$ and we take the temperature profile from the surface to its depth multiplied by two, in this way, we reduce the data from the deep layer, but making sure not to exclude

the isothermal layer or the thermocline. The sigmoid function presents central symmetry with respect to its inflection point, from this point, in both directions, the sigmoid presents a diagonal line, a curve and a straight line. Given these characteristics, by fitting the sigmoid function, we seek to fully represent the mixed layer with a straight line, locate the inflection point in the center of the thermocline and consequently represent the thermocline with the diagonal line.

First, we evaluate the direction of the vertical temperature change. To do this, we compare the temperature value near the surface against the deeper one, if the value closest to the surface is greater, the profile decreases with depth, otherwise it increases. If the temperature decreases with depth, the sigmoid function is inverted by multiplying it by $-1$, then we normalize the temperature data between $0$ and $1$.

Next, nonlinear least squares is used to fit the function to obtain the optimal values of the parameters $a$ and $b$. Once these parameters are obtained, it is possible to approximate the temperature values at any depth above the sigmoid. Despite the central symmetry that the sigmoid function presents, the nonlinear fit of least squares allows the fit to place one straight line shorter than the other one (without losing its shape), thus losing the symmetry and placing the inflection point in the center of the thermocline, regardless of whether or not it coincides with the greatest value of $N_T^2$. We assess the goodness of the fit with the coefficient of determination ($R^2$), this coefficient informs on how well the adjusted function approximates the real data, being 1 the best adjustment.

Once the sigmoid has been fitted to the temperature profile, we can determine the MLD and MTD by scrolling through the function. The temperature at a depth of 10 m resulting from the adjustment of the function is taken as a reference and is denormalized, that is, it is transformed again to be represented as a function of depth. The MLD is then determined as the depth where the potential temperature is $0.2\ °C$ higher (or lower) than the reference temperature at 10 m (de Boyer Montégut et al., 2004). To locate the MTD, we used the same procedure but going upwards in the function, in this case we take the reference temperature where the deep layer should be located and we look for the difference of $0.2\ °C$ by decreasing the depth through the function. Because the method is based on a single nonlinear function adjustment, we can have a precision of even centimeters. The procedure explained above can be seen in Fig. 1 and can be used through the script developed (Romero et al., 2022). To visualize the profiles of Fig. 1 up to 2000 m depth, see Fig. S1 of the Supplementary information.

This methodology was applied to each of the DMQC Argo profiles and consequently we provide the monthly average of $R^2$ in a $2°\text{x}2°$ grid as a proxy to know the regions of the ocean where the proposed methodology is reliable.

We have validated the method by comparing our results for the MLD with other existing methods. To do so, the MLD of each profile of the data set was calculated in four different ways: (i) with the proposed method, (ii) following the methodology of Holte and Talley (2009) and with the methodology of de Boyer Montégut et al. (2004), using both the (iii) density threshold and the (iv) temperature threshold (hereinafter, we will refer to the former three methods as HT09, B04D and B04T respectively). HT09 performs an evaluation of several criteria (calculated from temperature, salinity and density separately) to determine the MLD for each profile, while B04D use a threshold of $0.03\ \text{kg m}^{-3}$ compared to the reference value at 10 m depth in the density profile and B04T an absolute difference of $0.2\ °C$ compared to the reference value at the same depth but in the temperature profile, to locate the MLD. To compare the four methods to compute the MLD, the ocean was divided in regions following the reference of the Working Group I contribution to the Sixth Assessment Report (AR6-WGI) (Iturbide et al., 2020) of the

Intergovernmental Panel on Climate Change (IPCC) and the regional monthly average of the MLD was calculated. Regions with less than 10 averaged values were not taken into account. Finally, the Spearman correlation was calculated between the results of the four methodologies.

To carry out our computations we averaged all profiles available for each climatological month in $2°$x$2°$ cells. The choice of the $2°$x$2°$ cells responds to a compromise between keeping reasonable resolution and enough data in each cell for each climatological month. With these data, we then obtained climatologies of the MLD produced by each methodology described above including the one proposed here. Once the calculation of the MLD was validated, the monthly climatologies of the MTD, the thickness and the strength of the thermocline were obtained. The thermocline strength was calculated using the thermocline strength index (TSI), defined as $\Delta\Theta\,\Delta h^{-1}$ ($°$C m$^{-1}$), where $\Delta\Theta$ and $\Delta h$ are the differences in temperature and depth, between the MLD and MTD (Yu et al., 2010).

## 4  Results

This methodology developed here was applied to each of the DMQC Argo profiles marked as good or probably good data. As a preliminary assessment of the adequacy of the sigmoid function fit, we performed a first visual scan of random temperature profiles at different ocean latitudes. Figure 1 shows, illustratively, some examples of different temperature profile adjustment situations with different characteristics and geographical locations, where the MTD and the MLD computed with the method proposed here are indicated.

The temperature profiles shown in Fig. 1a-e were taken from high latitudes in the southern hemisphere to high latitudes in the northern hemisphere, while the profiles in Fig. 1f-j are located in regions where thick BL are found (see de Boyer Montégut et al. (2007)). In the profiles from Fig. 1a-e the temperature drops in the thermocline as the depth increases, while the profiles in Fig. 1f and h-j show temperature increase with depth. In both cases our methodology seems to accurately determine the MTD. Profiles in Fig. 1f and g show the greatest variability in $N_T^2$, but the quality of our adjustment differs between them. In panel f, despite the high variability in the deep layer, the methodology correctly determines the MLD and MTD. However, in panel g, high variability occurs from the end of the isothermal layer, and our methodology cannot perform the adjustment of the function correctly. In the same way, the B04T, $D_\sigma$ and VRI methods failed to correctly locate the MLD and the thermocline (as shown in Fig. S1). To illustrate the precision of our method and to identify regions where it should be applied with caution due to the variability of the temperature profiles, we provide a map of the monthly average of $R^2$ (Fig. 2).

In general terms, the adjustment of the sigmoid function is very good (with $R^2 \geq 0.9$) in low and mid latitudes. However, the cells with red and gray colors should be taken with caution. These present $R^2 < 0.3$ and $< 0.7$ respectively, which indicates that the adjustment of the sigmoid function was poor or not optimal. The worst adjustments correspond to the core of the Antarctic Circumpolar Current in the Southern Ocean, the North Pacific and the Western North Atlantic. These are regions where the stratification of the water column is dominated by salinity, there are temperature inversions and/or present strong currents and associated turbulent dynamics. In general terms, in the regions where the adjustment was worse, it was less good in winter months.

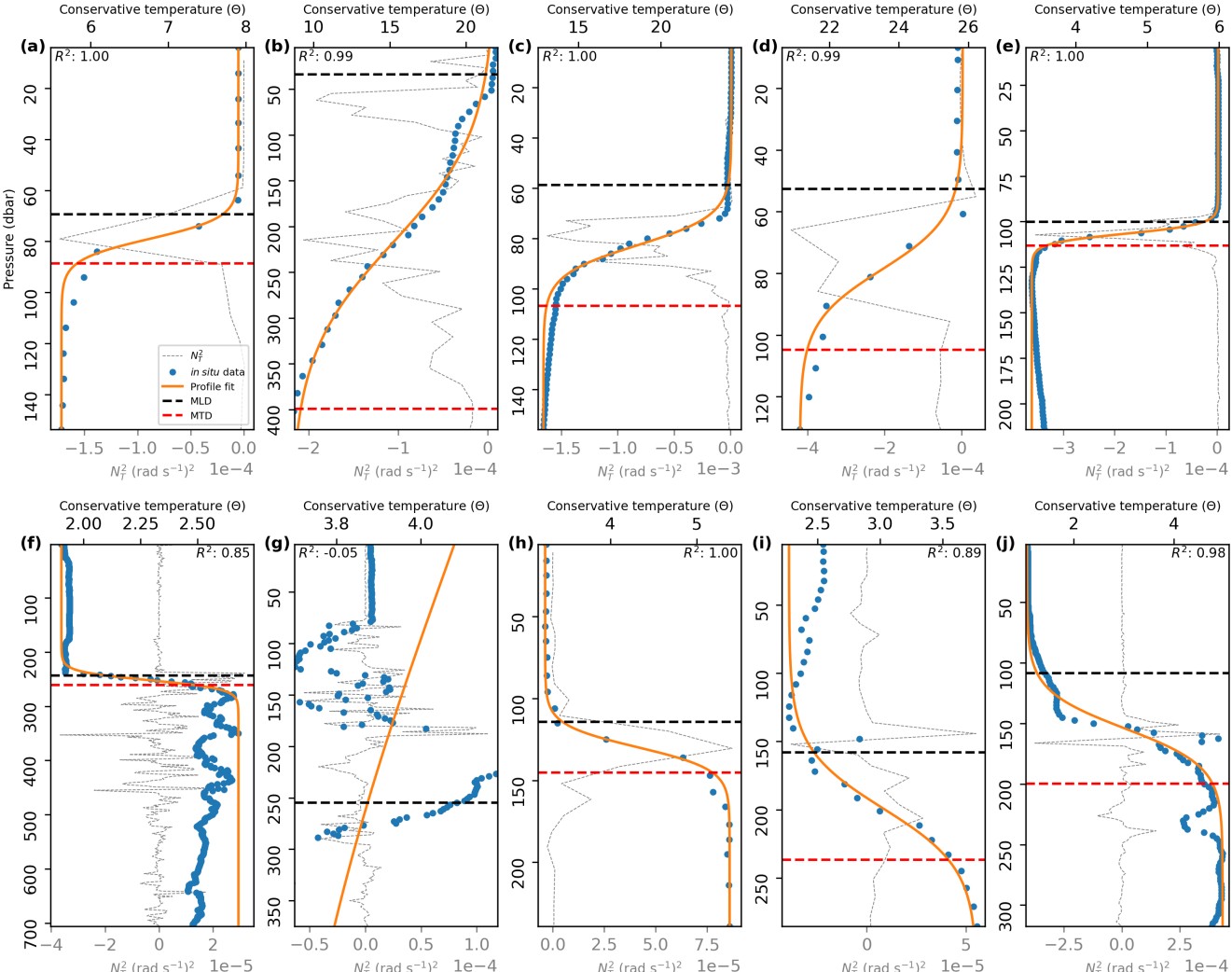

**Figure 1.** Location of the MTD (red dashed line) and the MLD (black dashed line) in temperature profiles (blue dots). (a) 52.95° S and 90.05° W on 23 January 2003. (b) 25.13° S and 93.47° W on 12 January 2013. (c) 1.90° S and 126.07° W on 25 August 2013. (d) 20.02° N and 41.14° W on 15 December 2015. (e) 49.00° N and 174.69° W on 13 December 2017. (f) 60.00° S and 116.86° W on 12 August 2015. (g) 55.42° S and 162.63° W on 12 August 2020. (h) 63.23° N and 54.20° W on 08 February 2010. (i) 56.07° N and 174.91° W on 20 February 2014. (j) 61.84° N and 54.27° W on 01 February 2016. Goodness of fit is shown at the top of each profile with $R^2$

The results of the preliminary evaluation showed that both the visual examination (not shown) and the values of $R^2$ indicate that our methodology correctly locates the MTD and the MLD at different latitudes. After this first step, we carried out the validation against other methodologies.

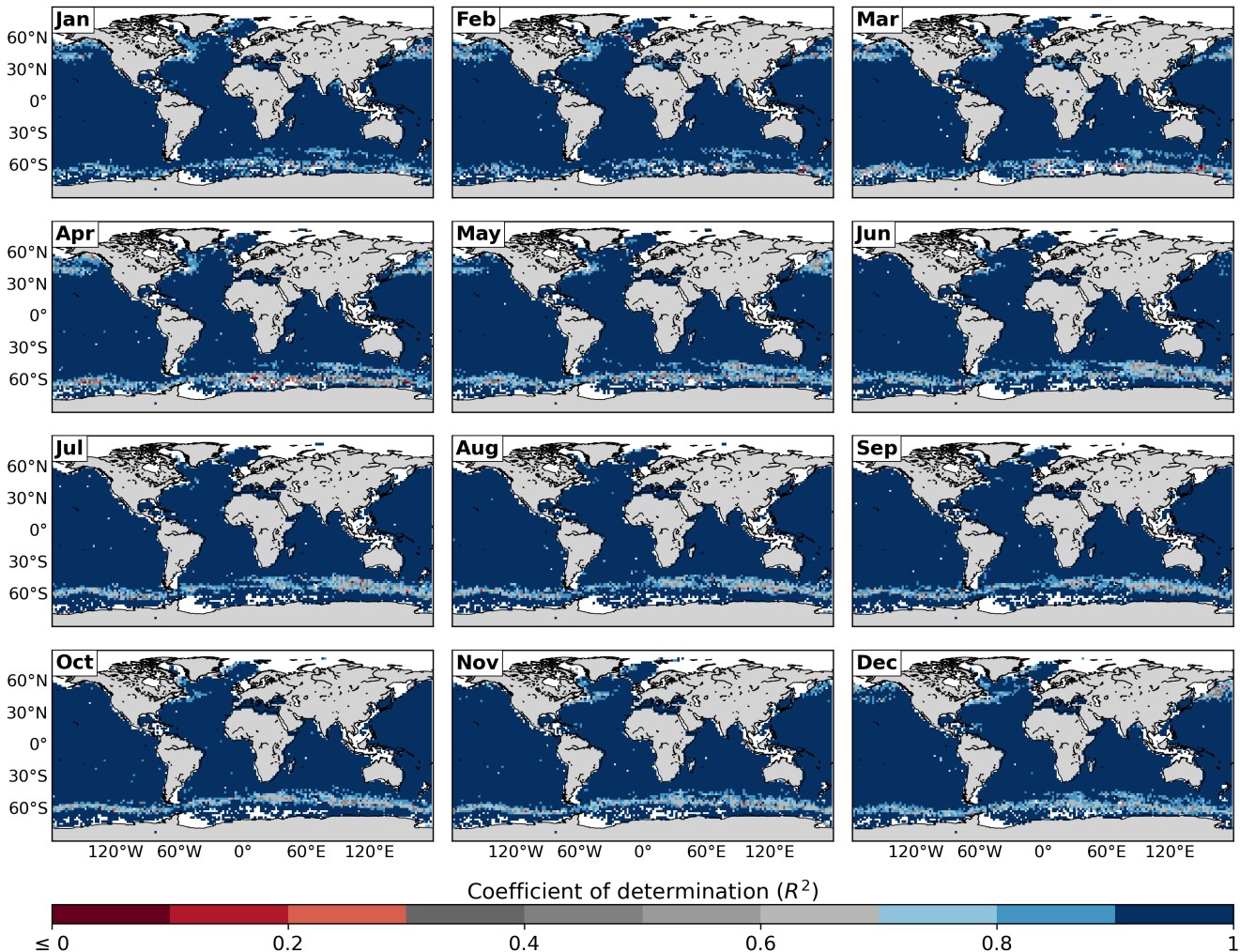

**Figure 2.** Monthly average of $R^2$.

## 4.1 Atlas of the mixed layer depth

The monthly climatology of the MLD computed with the proposed method (Fig. 3) reproduces well the spatial patterns and the seasonal variability of the mixed layer as shown in previous studies (e.g., de Boyer Montégut et al., 2004; Holte et al., 2017). It captures the regions with the deepest (northern North Atlantic and Southern Ocean) and the shallowest values (tropical and subtropical areas of both hemispheres) and their magnitudes.

The MLD shows strong seasonality as well as hemispheric asymmetry, mainly in the subtropical and subpolar regions. In the northern hemisphere, in summer months, the mixed layer is generally shallower than 50 m; while in late winter, it reaches climatological mean values over 1000 m in some regions of the North Atlantic basin such as the Labrador Sea, the Nordic Seas.

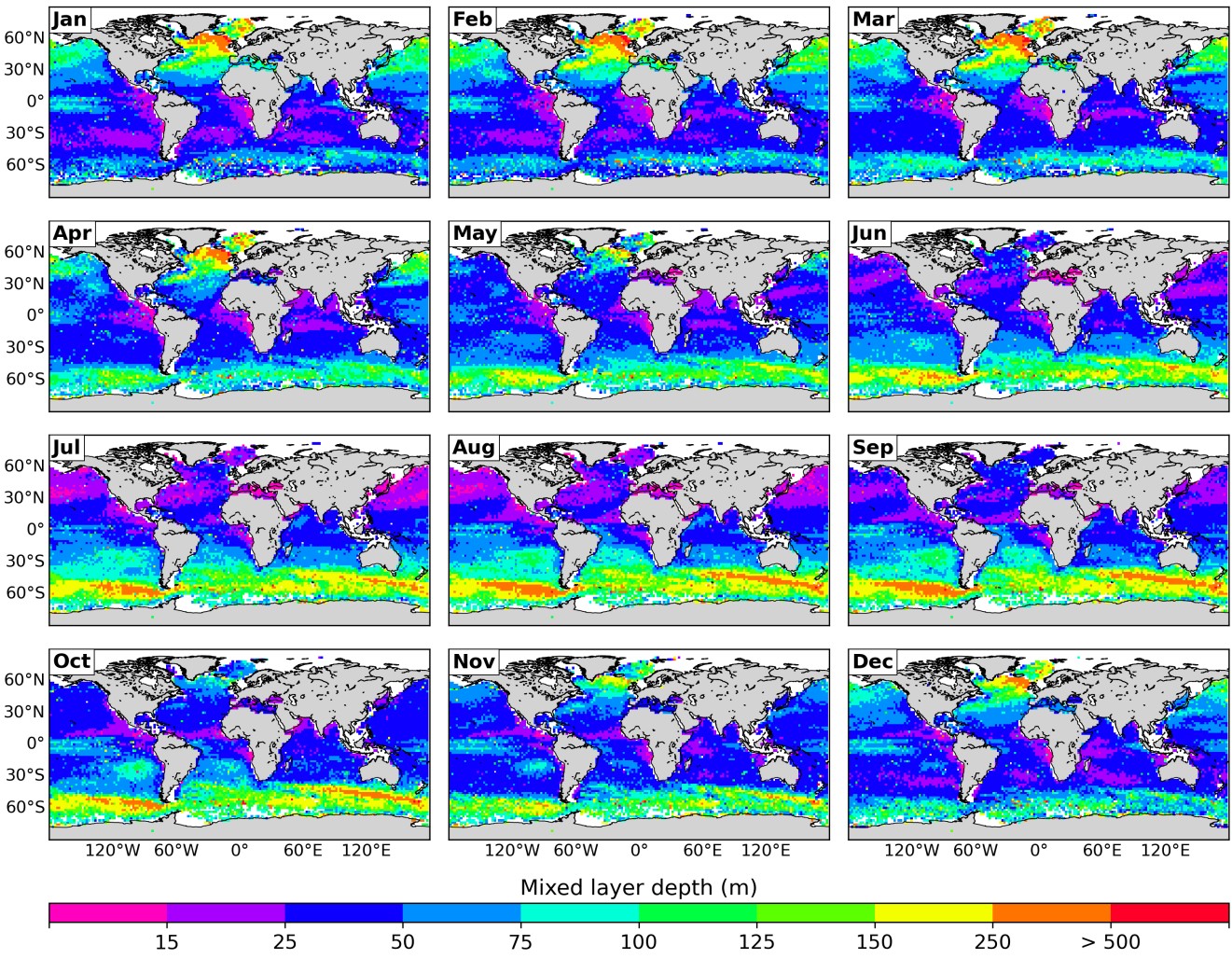

**Figure 3.** Climatology of the MLD estimated from individual profiles.

In the Southern hemisphere, the MLD is generally deeper than in the northern hemisphere, and it is dominated by the signal of the Antarctic Circumpolar Current. The mixed layer in this region varies between 75-100 m depth in summer and around 500 m depth in winter, mainly in the Indian and Pacific basins. In tropical and subtropical latitudes, the MLD is generally very shallow, varying below 15 m in summer up to 150 m depth in winter.

In general terms, our climatology agrees with those of Holte et al. (2017), de Boyer Montégut et al. (2004), HT09, B04D and B04T (Fig. S2-S5 in the Supplementary information), but there are some differences. The comparison between our method and HT09 (Fig. S2) shows no net underestimation or overestimation and the relative difference between the two methods is less than |25 %| over most of the ocean. The HT09 method gives slightly shallower mixed layers than our method in subpolar latitudes and deeper in tropical and subtropical regions. On the other hand, B04D and B04T generally overestimate the MLD with

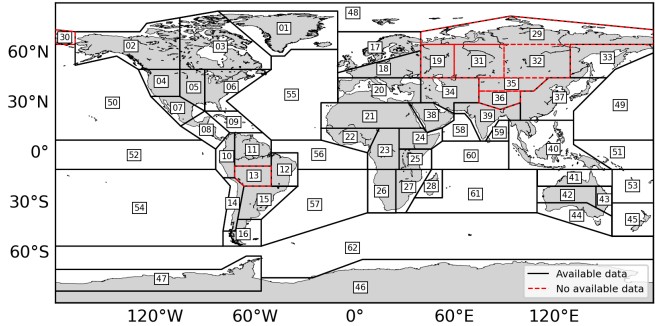

**Figure 4.** Reference regions of AR6-WGI. The regions used for comparison are those delimited in black.

respect to our method (Fig. S2 and S3). B04D has the greatest differences (above $50\%$) compared to our method, especially in the northern hemisphere from July to September, while B04T presents its greatest differences in tropical and subtropical latitudes throughout the year. To make a more quantitative comparison, the MLDs computed with the four different methods were averaged within the reference regions of AR6-WGI, as these were designed for regional synthesis (Fig. 4).

The red-delimited regions are fully continental regions and were not used for our analysis. The regions with less than 10 averaged values were also excluded. The averages of the MLD computed with each method in each region were plotted for a representative month of each season (Fig. 5).

     The MLD shows good agreement between the four methods in most regions. In general terms, the method proposed here is in better agreement with HT09 and B04T, with B04D the one that exhibits the greatest differences. In February and May,
the B04D method seems to overestimate the MLD in the regions of Northeast North-America and the Arctic-Ocean (regions 03 and 48, Fig. 5a and d). These are polar regions containing semi enclosed seas in the Northern Hemisphere. It is likely that these regions exhibit particular dynamics that complicate the detection of the MLD with a global threshold based method. One possible explanation is that coastal regions generally are worse sampled, but also, that the computation of the MLD can be complicated by a number of coastal processes such as river discharges or shallow bathymetry among others. Moreover,
the traditional delta density criterion of $0.03$ kg m$^{-3}$ has been suggested to underestimate the MLD in polar regions, as demonstrated in the study by Peralta-Ferriz and Woodgate (2015), where it was found that a better criterion for these regions is $0.1$ kg m$^{-3}$. Interestingly, the agreement between the four methods is also good in the regions where our adjustment was not considered to be good ($R^2 < 0.7$, Fig. 2). In other complicated region as the one around Greenland (region 01) there are some differences between the four methods, but the one proposed here either agrees with B04T in February and HT09 in August,
and gives an intermediate MLD value with respect to two other methods (as in May and November). The Spearman correlation between the results was calculated (Fig. S5), and showed high correlation between the results of all the methodologies, with the results of HT09 and B04T being the most correlated ($0.98$), followed by the proposed method and B04T ($0.95$). Finally, all the correlations between B04D and the rest of the methodologies showed values close to $0.90$.

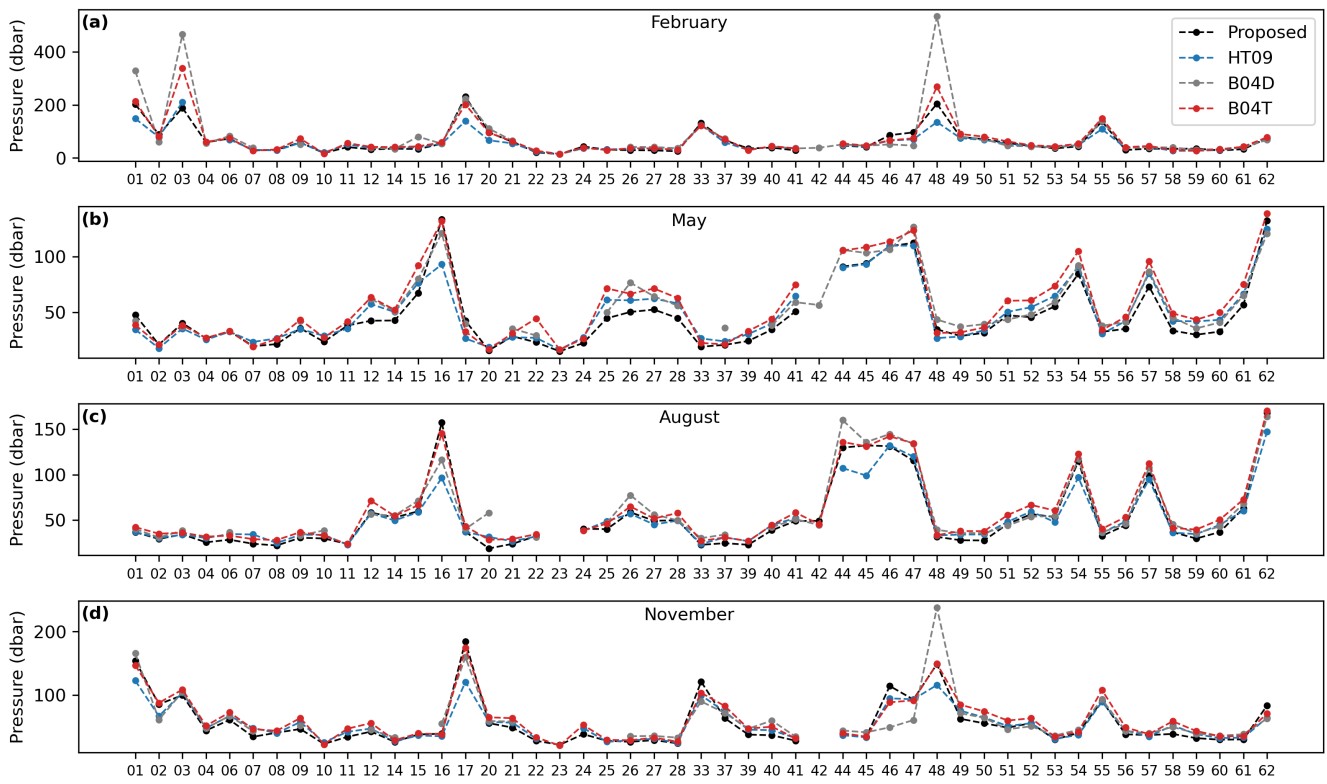

**Figure 5.** Comparison of methodologies to locate the MLD in (a) February, (b) May, (c) August and (d) November.

All this suggests that although our method is not perfect in highly dynamical regions, it gives results that compare well with other broadly used methods to detect the isothermal layer and the MLD, even in salinity-dominated regions. Moreover, this highlights the possible deficiency of all existing methods in detecting the MLD in these regions.

## 4.2 Climatology of the maximum thermocline depth, thickness and strength

Once the proposed methodology is validated with the calculation of the MLD, we computed the monthly climatology of the MTD (Fig. 6).

As expected, the shape of the MTD follows that of the MLD but the hemispheric asymmetry is not so evident. The subtropical and subpolar regions of the North Atlantic as well as the Southern Ocean exhibit the deepest thermoclines of the ocean. Similarly to the MLD, the deepest thermoclines are found in late winter: March-April in the Northern Hemisphere and September-October in the Southern Ocean. In the northern hemisphere, during the summer the MTD is generally no deeper than 100 m; while in winter, it reaches depths greater than 1000 m in the same regions where the MLD reaches its maximum values (Fig. 3), extending to the Gulf Stream in winter and spring months. In the southern hemisphere, the mean climatological MTD for the summer and winter months is similar to those of the opposite hemisphere. As in the case of the MLD, MLDs

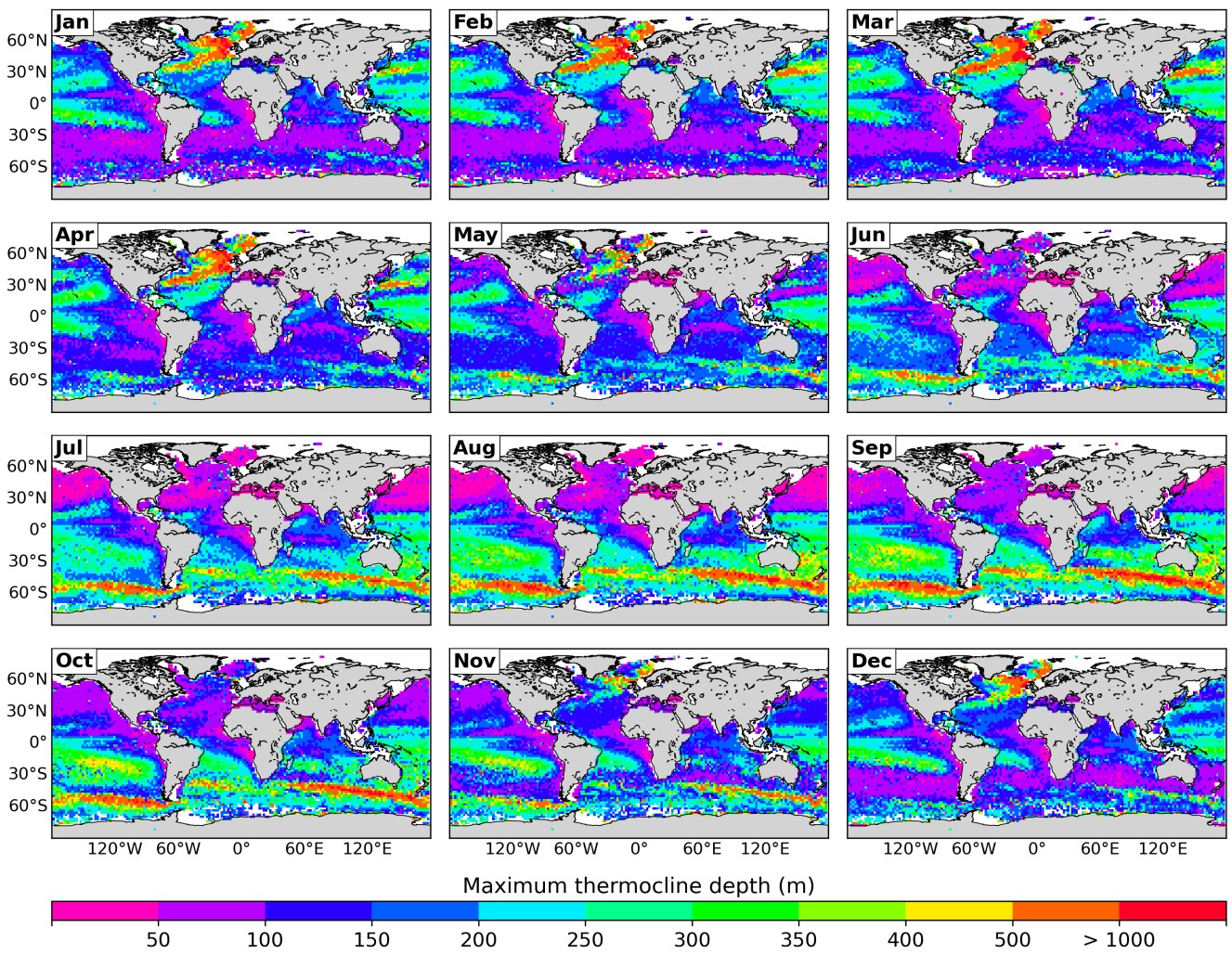

**Figure 6.** Climatology of the MTD estimated from individual profiles.

are relatively deep in the Antarctic Circumpolar Current in the Southern Ocean. During winter and spring, the MTD reaches values deeper than 500 m in its core, reaching more than 1000 m in localized areas. The deeper thermoclines in the Southern Ocean, in winter, coincide with the seasonality of the zonal band where the inertial horizontal kinetic energy in the mixed layer is larger (Flexas et al., 2019). This energy is injected by the relatively strong winds during this season, which is also related to the relatively deep MLD (as shown in Fig. 3).

During summer, the MTD rarely exceeds 50 m depth in the Northern and 75 m in the Southern Hemisphere. In these same regions, the climatologies of the thermocline thickness (Fig. 7) presented mainly values below 50 m.

The thermocline thickness follows a similar pattern as the MLD and the MTD. As for the MLD and MTD (Fig. 3 and 6 respectively), the climatology of the thermocline thickness (Fig. 7) presents a marked seasonality in subtropical and subpolar

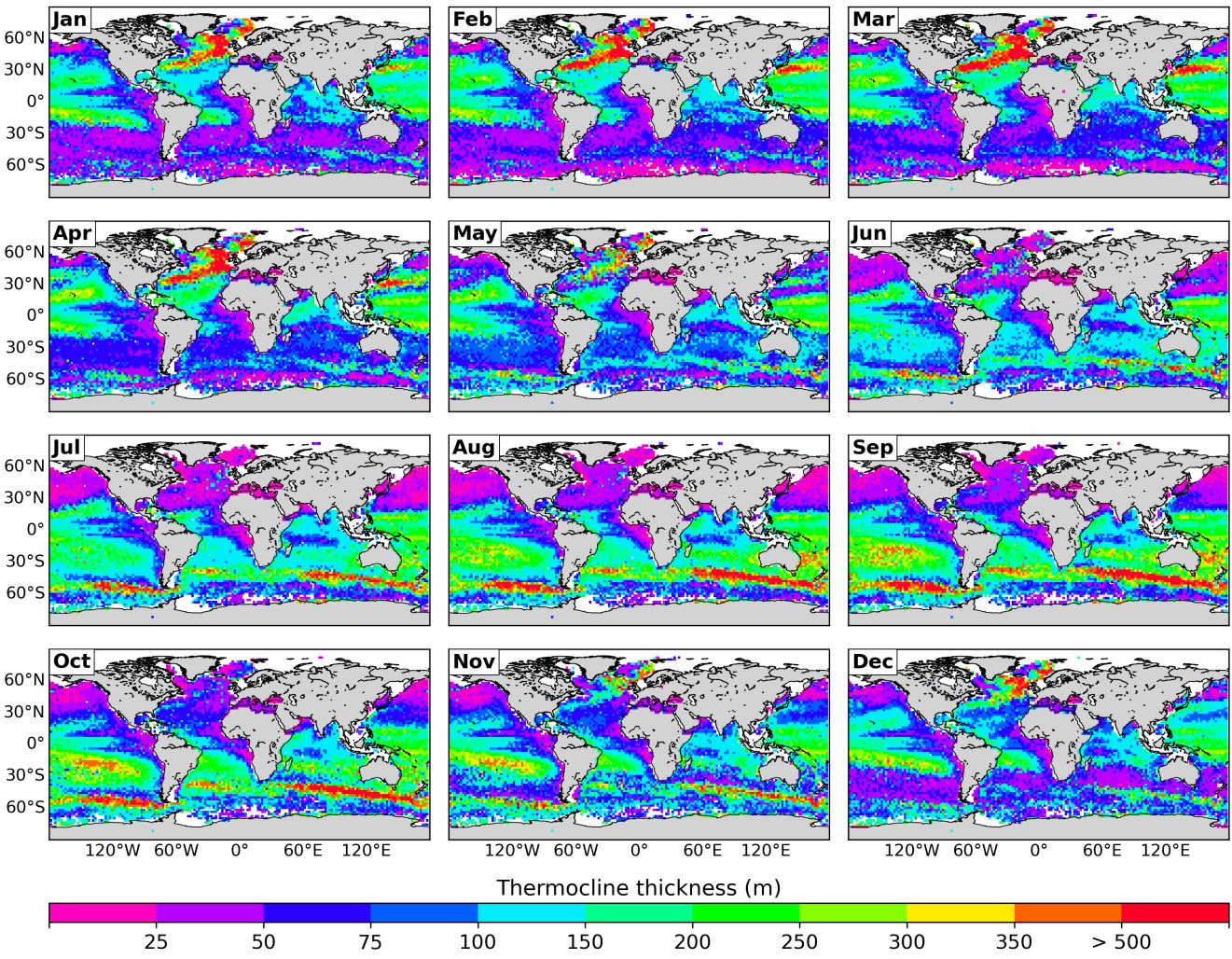

**Figure 7.** Climatology of the thermocline thickness estimated from individual profiles.

latitudes. As expected, the thickest thermoclines ($> 500$ m) are found in late winter and in the regions of lower stratification: March-April in the North Atlantic basin and September-October in the Southern Ocean, particularly in the region dominated by the Antarctic Circumpolar Current. Away from these regions, the seasonal variability of the thermocline thickness is low in tropical regions where it varies between 150 and 250 m depending on the region. The thinnest thermoclines are observed in summer in subtropical/subpolar latitudes of both hemispheres (July-August in the Northern Hemisphere and January-February in the Southern Hemisphere).

Finally, the climatology of the thermocline strength (Fig. 8) maintains the seasonality in subtropical and subpolar latitudes with a $|\text{TSI} < 0.1|$. On the other hand, the Tropical Eastern Pacific and Tropical Eastern Atlantic have TSI $> 0.1$ throughout the year, as do the North Pacific and North Atlantic, but from June-November. The Black Sea shows the highest values of

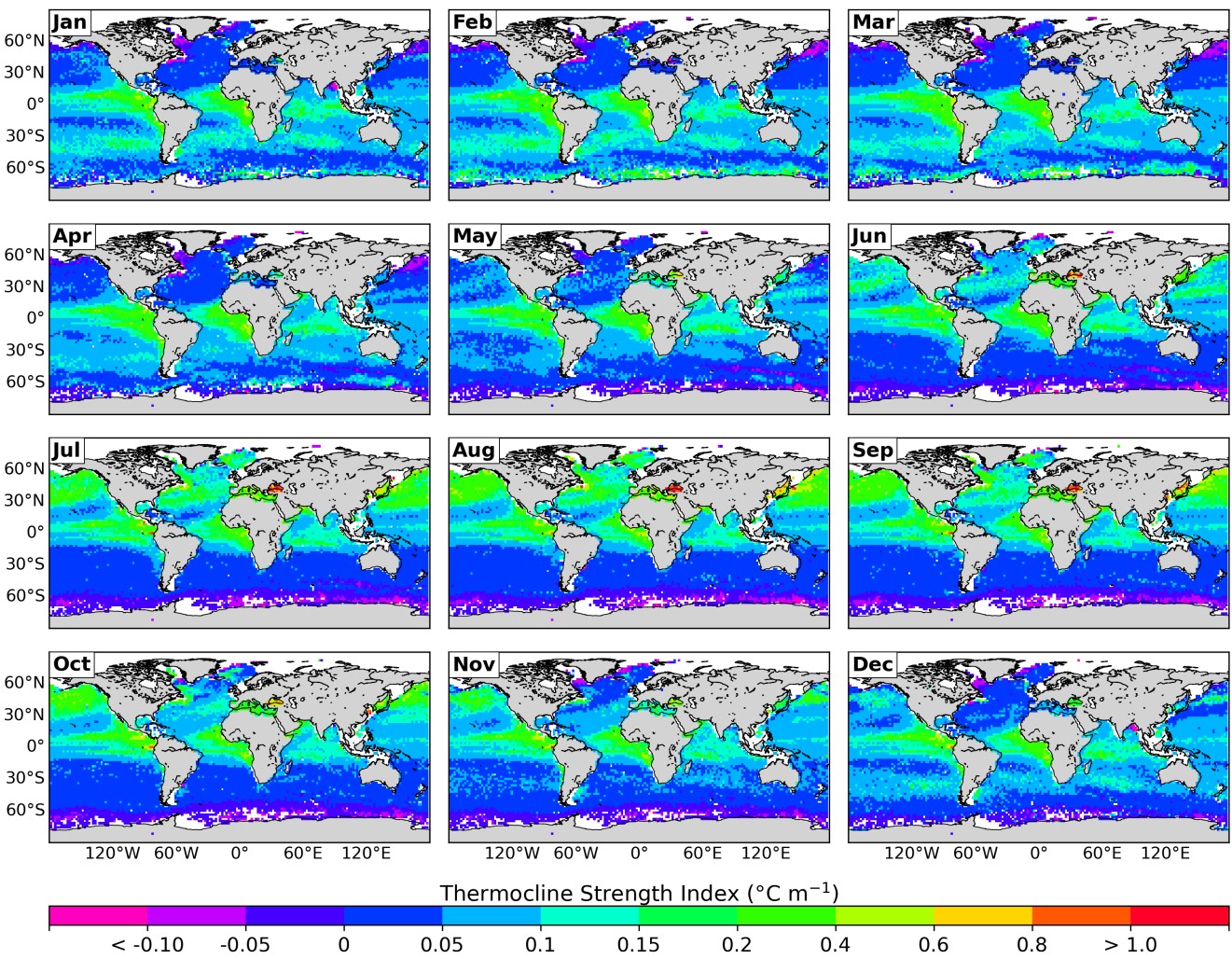

**Figure 8.** Climatology of the thermocline strength estimated from individual profiles.

TSI ($> 1$), while the lowest values ($< -0.1$) are scattered in subpolar regions and in the discharge region of the Ganges-Brahmaputra rivers (December-February).

## 5 Discussion

In this study we have proposed a new method to locate the strongest thermocline that lies just below the MLD of the water column. This method is based on the adjustment of a sigmoid function (in this case the logistic function) that relies on the principle that the thermal structure of the ocean consists of three main layers: the mixed layer, the thermocline and the deep layer of the ocean. Although not all temperature profiles have the s-shape mentioned above throughout the ocean, the proposed

method is based on the absolute maximum point of $N_T^2$ and the nonlinear least squares to fit the sigmoid function, to place the diagonal line of the function in the thermocline. Since the most stratified point of the temperature profile is used to place the sigmoid, the method locates the strongest thermocline which in most cases will coincide with the seasonal one.

The proposed method, due to the shape of a typical temperature profile in the ocean, also allows us to determine the MLD and, therefore, we were able to validate it. The climatology of the MLD generated in our study (Fig. 3), is in good agreement with those provided by de Boyer Montégut et al. (2004) and Holte et al. (2017). The four methods compared in this study (based on temperature and density thresholds), reproduce the magnitude, the spatial variability and the seasonal cycle of the MLD throughout the global ocean in a similar and consistent way for most of the ocean regions. This is because over most

of the ocean the stratification is dominated by the temperature (de Boyer Montégut et al., 2004). Moreover, in the few regions where all three methods disagreed, the one proposed here was mostly in line with the results of B04T and HT09, while B04D exhibited extreme MLD values. The latter is explained because the proposed method and B04T use the temperature profile and the same threshold to calculate the MLD. Moreover, HT09 uses a method that combines different thresholds, including a temperature-based one. The coincidences between the proposed method and HT09 can also be partly explained by the fact

that both methodologies use function adjustments. The study of HT09 bases on the homogeneity of the MLD to perform a linear function adjustment, in addition to performing another linear function adjustment in the thermocline, while the proposed methodology uses this same MLD feature to determine how deep the sigmoid function will be adjusted. The regions with the greatest differences between methodologies were found in a small region of the Arctic Ocean and the Labrador Sea (Fig. 5a and d). There, the method of B04D strongly differed from the others, we attribute this to the fact that staircase stratification has

been reported in these areas (Timmermans et al., 2008; Toole et al., 2011), defined as BL and CL regions (de Boyer Montégut et al., 2004). Indeed, our preliminary results have shown inverse thermoclines, multi-thermoclines and mixed thermoclines (Jiang et al., 2017) in these regions (not shown). In this sense, we have provided some examples of how our method could accurately locate the BL in some cases (Fig. S1 of the Supplementary information). In these cases, the MLD calculated by the 0.2 °C threshold (B04T) coincides with that calculated with our methodology. Conversely, in other cases, where the isothermal

layer was highly variable (Fig. S1g and i) our method was unable to locate the thermocline. The BL was also located to these profiles using the $D_\sigma$ and $D_{T-02}$ criteria defined in de Boyer Montégut et al. (2007), showing that when the thickness of the BL is of the order of hundreds of meters, the proposed method locates the inverse thermocline produced by the inversion of temperature that is found below the isothermal layer. On the contrary, in thinner BL, our method locates the thermocline below the lower limit of the BL. Sprintall and Tomczak (1992) define the BL as the distance that separates the MLD from the MTD,

however, when this BL is thick, the proposed methodology locates the inverse thermoclines that are within this barrier and not those that could be below $D_{T-02}$.

All previous studies compared here have used Argo data in their climatologies. However, de Boyer Montégut et al. (2004) being an older study had less data, while Holte et al. (2017) used real-time quality control (RTQC) data in a $1°x1°$ grid (in contrast to the $2°x2°$ grid used by us and de Boyer Montégut et al. (2004)). This smaller-size grid is not optimal to be used

with the amount of DMQC data available in this snapshot (Argo, 2022b), as about a third of cells would contain less than three values for monthly averages. Using RTQC data would increase the amount of data available for our computations. However,

Argo recommends using only DMQC for scientific research, since the RTQC tests are automated and may contain bad data, as explained in the manuals (Argo Data Management Team, 2022) and even using the best quality control flag, as shown in Romero et al. (2021), therefore, using this data could cause the erroneous computation of the MLD and MTD.

Using the density to estimate the MLD usually gives good results, since it depends on temperature and salinity. However, the density can show vertical compensation below the well-mixed layer (de Boyer Montégut et al., 2004) causing deeper MLDs calculated from density thresholds. Although B04D and HT09 methods use density profiles for the calculation of the MLD, this does not give good results with the methodology proposed in this paper. The adjustment of the sigmoid function bases on the typical shape of the temperature profile. In order to calculate the pycnocline with a similar methodology to the one

proposed here, it would be necessary to find another function that better fits the density profile, since despite also being able to be represented in three layers, the typical density profiles present an inclination along the entire profile, which makes it difficult to fit a conventional sigmoid function. In the polar and some subpolar regions of both hemispheres, where salinity is the major contributor to the density gradient it dominates the stratification. In these cases the thermocline and pycnocline may differ significantly, and this is why MLD calculations based on temperature profiles differ considerably from those based on

density in these regions.

     The calculation of the climatologies of the MTD, the thermocline thickness and thermocline strength were not compared with the calculation of any other method since no method of calculating these parameters was found that works on a global scale. Helber et al. (2012) mention that the transition layer thickness (TLT) used in their study may encompass the entire thickness of the thermocline, and in fact their TLT climatology presents some coincidences with our climatology of the thermocline

thickness (Fig. 7). The most notorious coincidences are in the regions of the northeast Pacific Ocean, the northern North Atlantic Ocean and the Antarctic Circumpolar Current in the Southern Ocean, where the thermocline thickness and the TLT show values greater than 350 m in the winter months, in addition to the marked seasonality that both presents in tropical and subtropical regions. Regarding the climatology of the thermocline strength (Fig. 8), it was calculated through the TSI, this index indicates the steepness of the thermocline (Duka et al., 2021). The further TSI is from 0, the slope is less steep and therefore

the thermocline is stronger. The strongest thermoclines found in the Black Sea are associated with thin thermoclines (15-20 m) between warm surface waters and cold intermediate waters ($20 - 8\,°C$) (Akpinar et al., 2017) which produce small slopes. Negative values of TSI are caused by inverse thermoclines, these were found mainly in subpolar regions and in the Ganges-Brahmaputra rivers discharge, these regions present TSI close to 0, which means steep slopes and therefore weak thermoclines. The formation of intermediate strength thermoclines (i.e. $0.2 < \text{TSI} < 0.8$) in the North Pacific and North Atlantic coincides

mainly with the months (July-September) when there are no BL in these regions (de Boyer Montégut et al., 2007). On the contrary, from January-March when the BL are thicker (de Boyer Montégut et al., 2007), weak thermoclines and regions with inverse thermoclines are shown.

     To compare the location of the MTD, the VRI method was applied to the profiles shown in Fig. S1 and only gave good results with those located in tropical latitudes. As shown in Fig. S1, far from the tropics, the calculation of the MLD and MTD

with the VRI method does not give good results, in addition to not considering the inverse thermoclines (Fig. S1f-j). For these reasons, the performance in fitting the sigmoid function were used to validate the method.

In different scientific areas, $R^2$ is used as a goodness-of-fit measure for sigmoid functions (e.g., Cao et al., 2019; Bhogal et al., 2014; Ritz and Spiess, 2008; Liu and Saint, 2002; Van der Graaf and Schoemaker, 1999), through this measure, our method showed generally good performance in the adjustment of the sigmoid function to the temperature profiles with the
exception of a few regions: the North Pacific Ocean, the northern North Atlantic Ocean, the Arctic Ocean and the core of the Antarctic Circumpolar Current in the Southern Ocean. While the interpretation of the MLD and the MTD in these regions has to be made with caution, it is noteworthy that the values of $R^2$ are not a direct indicator of the precision of the method to calculate the MLD and the MTD. Rather, this index shows the goodness of the sigmoid function fit to the temperature profile. Precisely, in the most problematic regions mentioned above, $R^2$ is lower than 0.7 in some months (Fig. 2). However,
the three methods used for the MLD calculation give very similar values (Fig. 5) even in these regions. This suggests that it is not a particular shortcoming of our adjustment. Previous studies such as those by Peralta-Ferriz and Woodgate (2015) and Pellichero et al. (2017) have shown the variability in the calculation of the MLD in these regions depending on the methods and thresholds used. This has evidenced that these are complex, highly dynamical regions (i.e., turbulent regions with important eddy activity), where the estimation of the MLD in a reliable way is a complicated task. In this sense, it has been suggested that
in the Southern Ocean the MLD calculation is less accurate than in regions at lower latitudes where the water column is strictly temperature stratified (Dong et al., 2008). The low values of $R^2$ shown in Fig. 2 are due to the abrupt changes in temperature in the profiles measured in these regions. These abrupt changes might be related to well known processes taking place in certain regions of the ocean as: (i) double-diffusive staircase stratification in the Arctic (Timmermans et al., 2008; Toole et al., 2011), and (ii) to the temperature inversions due to the influence of salinity that present different vertical structures in the Southern
Ocean (Dong et al., 2008).

The results of our adjustment also evidence the regions of the ocean where the water column exhibits a typical vertical thermal structure in three layers and the regions where, due to their dynamics, the structure of the water column cannot be divided in these three layers. The efficiency provided by the proposed method for the calculation of the MLD and the MTD, allows to perform local to global studies. For example, in the context of ocean warming, the differences of these layers could
be compared over different timescales to analyze the changes of the water column, detecting areas of the ocean where the thermocline has changed its depth, thickness or strength over time, and therefore to be a parameter of the potential effects on the pelagic ecosystem and socio-economic repercussions.

## 6   Conclusions

In this study, we present a methodology to locate the minimum and maximum depths of the strongest thermocline, its thickness
and its strength by adjusting the sigmoid function to the temperature profiles in the global ocean. This methodology can be applied in those areas of the ocean where the water column can be divided into three layers according to its thermal structure. Our methodology gave good results in its validation against other three broadly used methodologies in the global ocean. The MLD computed with the four methods showed a high correlation, even in regions where the coefficient of determination

suggested a poor adjustment. This suggests that it is not a particular shortcoming of our method, but rather a general difficulty
in determining the limits of the three typical oceanic thermal layers in highly turbulent regions.

*Code availability.* The methodology presented in this study was developed in Python 3.7 and is licensed under a Creative Commons Attribution 4.0 International License. The source code is available at https://doi.org/10.5281/zenodo.6985561 (Romero et al., 2022). The latest package version is v1.1.0.

*Data availability.* These data were collected and made freely available by the International Argo Program and the national programs that
contribute to it (https://argo.ucsd.edu, last access: January 2022; https://www.ocean-ops.org, last access: January 2022). The Argo Program is part of the Global Ocean Observing System. The data used belongs to the snapshot of January 2022 (Argo, 2022b).

*Author contributions.* ER developed the methodology described in this work and carried out the analyses. All co-authors contributed to the conceptualization and design of the study, the interpretation of the results, and the preparation of the article.

*Competing interests.* The authors declare that they have no conflict of interest.

*Acknowledgements.* We are grateful to CONACYT for granting scholarship no. 780669 to Emmanuel Romero. We appreciate that these data were collected and made freely available by the International Argo Program and the national programs that contribute to it (https://argo.ucsd.edu, last access: January 2022; https://www.ocean-ops.org, last access: January 2022). The Argo Program is part of the Global Ocean Observing System. We also thank the Centro Interdisciplinario de Ciencias Marinas (CICIMAR) for their institutional support. We also acknowledge the critical comments from the reviewers.

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
