# Peer review of "Improving the thermocline calculation over the global ocean"

_EGUsphere, 2022_

## Referee Comment (RC1)

Figure. Example ARGO data of no. 2902688_104 (date: 2019/07/26, position: 127.98°E, 15.65°N), (a) conservative temperature, (b) buoyancy frequency profile and (c) MLD and MTD calculated by the provided code.

---

## Referee Comment (RC2)

##################################################################################

Review of the manuscript:
"Improving the thermocline calculation over the global ocean"

Submitted to Ocean Science, September 2022
Review sent, 1st February 2023

##################################################################################

Summary:
========

This paper presents a new method to diagnose both the MLD and the thermocline characteristics (thickness, depth, strength), from observed ocean profiles of temperature and salinity. Based on Argo profiles, a new climatology of MLD and thermocline variables is constructed using this method. It is compared with previous similar climatologies of MLD to validate the approach, and then the results about the new thermocline fields (depth and thickness) are discussed.

General comments:
================

This work is innovative because it proposes a new method that can be applied to get both the MLD and the thermocline characteristics at the global scale for the first time. This topic of ocean stratification is quite important and had several noticable studies lately (eg Sallee et al 2021), because of its potential impact on climate evolution. For those reasons, it would make sense that this rather method/data paper could be published in OS. However, when reading the paper, many serious problematic points appeared in the presentation of the method and results, showing that there is somehow a lack of knowledge about the MLD/stratification problems (unprecise method presentation, no word at all about barrier layers, comparing different types/family of MLD for validation, no introduction/references/discussion of the 3 layer hypothesis, a quite weak discussion of the results...). After having hesitated to reject the paper I must say, I would still advice a major revision of this work because the topic is relevant and actual, even if I think that the task is big to come to a publishable version. I give below precise comments about the modifications I suggest. There are especially 4 major points (and part of those points may also be developped in the detailled comments section) that, for me, must be adressed seriously before any possible publication.

Major points to address:
====================

1) You should clarify the description of the method :
I have questions about the details of your method, which is the base/heart of the paper. It is very important that the method is presented clearly and univoquely, here it is not the case for me at least. There are several detailed points to address (next section of the review, fig1, eqn 1, l99-100, l143-144) about this hereafter. Here is how I see the process of your method from a T/S profile :
  - get the depth D1 of vertical max of N2 from the density profile
  - multiply D1 by two to get the depth D2 over which we make the coming fit (D2 = 2*D1)

- evaluate the direction of vertical T change --> how do you do this ? It looks not present in your python code (?) and it must be explained here, is it by computing the difference T(0)-T(D2) ? something else ?
- amend the sigmoid function sign according to previous test
- normalise the temperature data between 0 and 1
- the fit : non-linear adjustment of the sigmoid function $f = +/- (1/(1+\exp(-a*(x-b))))$, to find the best a and b parameters, along with a goodness of fit coeff $R^2$
- get MLD by threshold on the T sigmoid (0.2 degC from 10m), get also MTD by threshold of 0.2 degC from the deepest level value of our interval (i.e. at D2 right ?)

If this process is right, then what you basically do is to fit a sigmoid on the temperature profile. Then you will diagnose a temperature-MLD (or isothermal layer depth) + its thermocline, and not an isopycnal layer depth + pycnocline.

. Then I do not understand why you diagnose your working depth interval (ie D2) from the density profile, logically it could be done directly from the temperature profile by searching the max of grad(T), because it is the pattern you basically look for and on which you perform a fit ?

. Also by doing this, you will end up with a sigmoid that is not centered on the N2 max, i.e. your b parameter in the sigmoid fit will be different from zero, because your thermocline is rarely exactly at the depth of the N2-max (as shown in fig1f cf detailed comments). This is not what is in the text when saying l. 99 that you "locate the most stratified point in the center of the sigmoid function".

2) The area/conditions of validity of your method should be clarified, with possible evolution toward application on density profiles :

There are detailed comments below for lines 143-152. This part should be re-written I would advice, and the context of Barrier Layer should be presented as it is strongly linked to this part and it is never mentionned. Your method diagnose MLD from temperature profiles as presented here, so obviously, you will miss the density-MLD (B04 or HT09) in all BL areas, which are quite numerous despite what you say in several places of your text. But this is not a problem according to me. To get a correct density-MLD, you should simply apply your method also to the density profiles, which have also often a 3-layer shape (even more than temperature, and no density inversion occur while you may have temperature inversions). This may make a bigger paper, but as it is now I have wondered why you stick to thermocline only, trying to say that it is nearly same as pycnocline (which is not true in many areas), while it seems that you have everything to do the computation directly to get the pycnocline also. Then when you compare with previous climatologies, you should compare what is comparable, if you have a temperature-based MLD (as it is now) then you should compare with a temperature-based MLD also (eg a 0.2 degC threshold MLD compared to 10m). Otherwise you will end up discussing Barrier Layers (BL) and compensated layers issues which have been discussed previously in papers and may not be the goal of your work (which seems to validate your new dataset of MLD and Thermocline or pycnocline if ever).

3) As stated above, you should compare climatologies with at least same family of criterion :

Your criterion is a threshold criterion of 0.2 degC on a fitted temperature profile. You should then compare it to the 0.2 degC threshold criterion of B04 also on temperature profiles not to the 0.03 kg/m3 density criterion, or you will mix several source of differences, especially salinity effects (cf lines 175-182). B04 and HT09 are shallower than your temperature-MLD mostly in the BL areas and high latitudes, which makes sense. B04 is deeper than HT09 which also makes sense as HT09 is a mix of several criteria and takes the smaller one, and this has also been already noticed in HT09 paper. Here you want to validate your MLD method, not discuss where salinity plays or not, for me it is another problem that has been tackled in many Barrier layer papers already. So you should compare your temperature based MLD with a similar temperature based MLD (eg deboyer 2004 0.2 degC threshold). Then if you want to have a part to see where your temperature MLD is biased compared

to a density MLD, you should first cite the BL papers, and then make a comparison that should well correspond to BL areas as is seen in your figures already.

4) An extended discussion about what you expect to diagnose as your thermocline is necessary, along with presenting in introduction what you talk about with this 3 layer hypothesis. Will you give the permanent thermocline or the seasonal one ? is it just below the MLD or not ? is it a mix of those 2 features ? (see helber et al JGR 2012, sprintall cronin EOS2001, Johnston Rudnick JPO 2009...)

Other detailed points to address :
============================

l. 22-23 : the ocean surface layer density and extent of the MLD is also determined by salinity in all "Barrier Layers" area which are present all over the tropical ocean (e.g. Sprintall and Tomczak JGR 1992, de Boyer Montegut et al JGR 2007). This should be mentionned here, as the total areas where salinity drives the stratification and MLD is actually not small at all (maybe about 20% of the total ocean surface when checking maps visually in the 2 references above).

l. 26 : "The characteristics and ... (coast-ocean gradients)" --> I think this sentence does not bring much information actually but could be kept as a transition. However, I would have said "spatially" instead of "latitudinally and longitudinally". Also, I do not really get the meaning of the term "coast-ocean gradients" here. It may be worth explaining further or being more precise, or else remove it.

l. 29 : I advice the change: "heat flux" --> "net heat flux"

l. 59 : "MTD" : I could not find the definition for this before (guess it is for "Maximum Thermocline Depth")

l. 62 : I am surprised that I could not find any trace of the Jiang et al 2016 publication via google, may be my mistake, but talking about maximum curvature method, I also though of the paper by Lorbacher et al JGR 2006, and would have liked to see how the Jiang paper would compare to that one, with the possibility of citing also the Lorbacher paper as being before in time, if it suits the situation of course.

l. 79 : you could also tell what quality control flag you use for selecting the acceptable DMQC data, only the flag 1 ? both flag 1 and 2 ? (of course you removed the 3 and 4 QC flags data I suppose)

Figure 1 (and equation (1) of sigmoid):
- from the description of your algorithm, you multiply the depth of density stratification maximum by 2 to get the max depth on which you fit you sigmoid, but on this figure, you do not show profile on this complete depth in (a) to (d) panels (e and f look quite ok to me). For exemple, in panel (a), the max stratif N2 (center of sigmoid) appears to be at 80 meters depth, so we could have expected that you show the profile on the whole depth of the sigmoid fit (ie 160m, but you stop your plot around 140m), and same in b,c, and d. Is there a reason for that ? if not, I think it could be somehow better to see the whole depth on which the fit occurs.
- in panels b and f at least, it looks like the max of N2 is not the center of the sigmoid, with the limits of the thermocline you diagnose being even below this depth fro panel f. This is apparently in contradiction with the algorithm you describe, or I may have missed something (as said in your texte you multiply by two the N2 max depth so that your sigmoid is centered at the max of N2, no ?). This may show the difficulty of mixing a density diagnostic (max of N2) with a temperature one (fit on T profile) because the 2 are not always linked, and their maximum gradient can often be at a different

depth. This may also be linked to the fact that your actual sigmoid is not necessarily symetric with the max of N2, which is necessary because you fit on T-profile, not on density-profile. Indeed, if I believe the formula found in the python code of Romero 2022, it is "1/(1+exp(-a*(x-b)))", so you should give this formula in your equation (1) so that readers know exactly what variables you fit (ie a and b here), and that your final sigmoid is not necessarily centered on the max of N2, but centered on the max of grad(T) as you make the fit on T, not on density.
- as an illustration, I would advice to also show a typical exemple of a profile when your method does not work so well as it happens quite in several extended places, i.e. when the 3-layer shape is not valid for temperature (R2 below 0.7 or 0.5), this should happen a lot in high latitudes beta-ocean with many interleaving features and several temperature inversions, or maybe in cases of remnant layers when restratification occurs by steps during spring at mid latitudes.

l. 119 : "HT03" should be "HT09" I think

l. 120 : here it would be better to (re)precise also the reference depth of B04 as the scope of the paper is about methods to diagnose MLD and stratifications, e.g. "... a threshold of 0.03 kg.m-3 compared to the reference value at 10 m depth in the density..."

l. 121 : "...de MLD..." should be "...the MLD..."

l. 136-137 : from your figure legend lon/lat data, I have profiles a, b and f in southern hemisphere, and d and e in northern, so you should correct something somewhere to be sure where we are.

l. 140-141 : Your conclusion is somehow a scientific overstatement. You cannot draw a general conclusion from a sample of 6 profiles among 2 millions or more. This analysis is not scientifically reproductible. If you take another 6 profiles, you may have a completely different conclusion. So it cannot be used for a general conclusion. Such exemples of profiles are nevertheless important to be shown so that the reader can realize some of the main details of the method illustrated in different situations (for exemple the fact that in austral the max of N2 is not at same depth as the max of thermocline, cf remark above for figure 1 also). I would advice that you temper your conclusion saying that this figure is an illustration of how the method works, and of the fact that it seems to work well (but this will be quantified afterward in the paper actually).

l. 143-144 : Again, for me this statement is not exact or misleading (and same at l.99-100). From fig 1. and formula in your python code, we can see that your sigmoid is not necessarily symetric with the max of N2 being the central point of the sigmoid (not central if b param is not zero in your sigmoid, cf also major point to address). Eventually I do not see exactly why you need to show that your method should be applied with caution in salinity dominated region. This is basically obvious from your method : you diagnose the isothermal layer depth (or temperature-MLD) because you work on the T-profile only, so of course in every Barrier Layer areas, you should end up with a too deep temperature-MLD compared to a density-MLD, and this may happen in all high latitude areas (beta oceans) and all tropical and mid latitude Barrier Layers areas (cf maps by sprintall tomczak JGR 1992 or deBoyer Montegut et al JGR 2007). Eventually your method is another way to compute the MLD (here based on temperature, but it could/should have been applied on density also) with one advantage that is to give also an estimate of thermocline depth and thickness which is a nice plus of your method of course.

Figure 2 and l. 144-152 : Here you show the T/S contributions to the stratification at the depth of the max-N2 (max density stratification). If we assume that this depth is most of the time at the base of the isopycnal mixed layer, then this analysis is very similar to the Barrier Layer (BL) mapsmentionned above. It is then surprising that all the historical BL in western equatorial paicifc are not more pronounced, but still they are present in your plot and should be noticed (As your map is not

seasonal, only the permanent BL patterns are well evidenced, or as the diagnostic is not really the same there are small differences, but in fig 8 the shaded areas indeed occur in west equat pacif during jun to sept for exemple.)

l. 151-152 : This statement is misleading to me. Your present diagnostic does not tell you about the vertical thermal structure of the ocean. You may have a BL situation typically in the west equat pacif, where you have warm waters at the surface then a thermocline with decreasing temperature at about 60m and colder water below (a thermal classic three layer structure). Yet you also have fresh water eg in the top 20 or 30 meters, with a 1st pycnocline at this depth, dominated by salinity. What figure 2 tells us is just what happens at the depth of N2 max. This is rather the R2 that tells you if your T-profile is well fitted to a sigmoid and hence if it has a 3-layer shape. Eventually I wonder about the relevance and this figure in your work, or it needs to be better presented, within the context of BL I would say, and may be seasonally.

l. 193-195 : I do not really agree with this statement, and this is not what was shown in HT09 and from your map in fig S1c, where B04 density MLD is shown to over estimate the MLD in high latitudes. This also makes sense with the fact that the ocean has a very low vertical stratification at high latitude, which pushes for smaller density criterion to get the mixed layer, especially in winter. And even if Perralta-Ferriz and Woodgate 2015 used 0.1 kg/m3 (but based on a heuristic method for choice), they recognize that they had to use a smaller 0.03 kg/m3 for several of their winter profiles to avoid over estimation of MLD.
Also you show that B04 overestimate MLD in arctic ocean in winter (region 48 of fig 6 a b), and then at line193 you say that their 0.03 criterion underestimate the MLD in polar regions. This makes not sense to me.

Figure 6 : I advice to replace this one and fig S1 by a plot of the maps of differences between HT09/B04 and your method, at 2 relevant seasons at least, which makes 4 maps, and enables a more detailled analysis than the AR6-WGI regions (why choosing those?), that were not designed for MLD studies and are not really relevant for that (for exemple dividing the labrador sea deep convection region in 2 parts)

l. 204-205 : As a remark here, I would say again that your MLD method validation is not the main point fo your article. Basicaly, your method for MLD diagnostic is the 0.2 degC threshold on a temperature profiles (after a sigmoid fit, that is your plus so as to diagnose thermocline after), so it is nothing really new for me, and it should be validated with a previous climatology of the same type (a threshold of 0.2 degC from T profiles), and should work with no big surprise to me. The important and new part of your paper is the thermocline (or pycnocline if you add it) part that comes now indeed.

Figure 7 : In may an extended pattern of MTD reaches 350 to 400 m in north atlantic, while it is about 100m or less in MLD (linked to the about 300 m thickness in figure 8 also). At that time restratification already started form the MLD shift from April to May, then I wonder why you do not catch this restratification with your method. I would have expect to see the MTD not so far from the MLD in may when the restartification signal is there in MLD, why is it not the case ? Are you diagnosing the stratification below the MLD (which I doubt actually) or a stratification that corresponds to the permanent thermocline ? You must talk about this somewhere, and also you should define your view of the three layer more precisely, because this view corresponds to warm surface / permanent thermocline / deep cold ocean, but with your method that limits the fit to max of N2 depth * 2 we would think that you may not catch the permanent thermocline but the seasonal one (or trnasition layer after Johnston and rudnick JPO 2009, see also Helber et al JGR 2012 that tackled also a related topic). For such a discussion you can also rely eg on Sprintall & Cronin EOS 2001 (upper ocean vertical structure) for a basis.

l. 213-218 : this description does not really bring something new to the paper, and explanations are quite well known already

l. 220-221 : same remark as above, here if you have very close values of MLD and MTD (either small or even large), then it means the difference MLD-MTD is small and based on your sigmoid that is symetric, 2*diff (which is your thickness) is also small, nothing special that is worth noting for me here, or I missed something.

l. 239 : your term "mixing" here is not correct. the mixed layer is something different than the mixing layer and you should be aware of that when writing a MLD/thermocline paper I suppose. See Brainerd and Gregg DSR 1995 "mixed and mixing layer depth" to know about this point and make the difference between the 2.

l. 259-266 : precise RTQC and DMQC i do not think it has been done before.
Also I doubt when you say "a large number of cells would contain less than 3 values for monthly averages". I do not have the same result on my side, I have a map with number of argo profiles per 1 degree boxes annually and it is over 50 over nearly the whole ocean, which means on average over 4 profiles per month at least. So I have difficulties to understand from where you make this statement.

l. 267-269: Your argument is not correct here. Oceanographers have pushed for salinity measurements historically because the latter also plays a role eg in MLD and now that we have those data (with argo), you say "it is better to not have it and just use temperature" and for that you rely on a very good but very old paper (rao 1989) at a time, especially in indian ocean, when they had no choice at all about using temperature and/or salinity for diagnosing MLD. This has nothing related with a scientific choice. Also if you rely on temperature only you must face the BL problem that is more extended than compensated layers, and about which you must talk somewhere.

L. 271-272 : This is not convincing me. For me it is the temperature profile that is more complicate to fit with a sigmoid, as it can have many features (interleaving in high latitude, several inversions, fossil/remnant layers etc...), while the denisty profile do have a simple chape as the ocean is stable, so light waters are always above heavy ones, and the density profile is the one that have by default the shape of the sigmoid. If it is not the case for you, then you need to show precisely why this is not true and where. As I said before, I would expect that you do your method on density profiles also, so I really do not understand why it cannot work for those. You must discuss about this with clear evidences of the fact that fitting T is easier than fitting density.

l. 282 : see Helber et al JGR2012, they study the startification below the MLD and they have fields of stratification below the MLD as I remember.

l. 283-284 : repetition of sentence, problem of edition to correct

Conclusion : it should be extended, especially talking more about the real new point of your article which, again, is not the MLd for me, but the thermocline (and hopefully pycnocline) characteristics that you estimate here.

To finish, you never show a map of the thermocline strength, while you diagnose it (as slope at the center of the sigmoid for exemple), why not ? I think this is a new interesting field/variable that must also be shown and discussed here as it gives the strength of the link between the ocean surface and interior, it would be a plus of your paper.

###############################################################################

---

## Author Response (AR1)

**Reply to Anonymous Referee #1**

Response to the reviewer comments

The Authors thank the reviewer for their comments that have helped to improve our manuscript. We hope that the reviewer finds our manuscript now suitable for publication in Ocean Science. Hereinafter, the reviewer's comments are in black and the authors' answers in blue.

**General comment:**

This paper proposed a new method for determining the upper and lower bounds of the thermocline. The method applies the sigmoid function to fit the Argo temperature profiles, then locates the mixed layer depth (MLD) and maximum thermocline depth (MTD) by using a temperature threshold (0.2â„ƒ ). The authors provided convincing evidence that the new method can determine similar MLD as the other two widely used methods (HT09 and B04). Next, they presented the global climatology of thermocline thickness and characterized their distribution patterns. The method is easy to conduct, and the MLD and MTD can be calculated as promised in the paper, although some details need to be reconsidered. The paper is well organized with a clear presentation, and I believe the new method is of great value to the readers and will help oceanographers who are interested in calculating MTD, after some adjustments.

**Major comment:**

The authors used R2 as the criterion for the goodness of fitting, while R2 values can be high even if the fitting functions don't fit the data well.

We thank the reviewer for bringing this up as the evaluation of the errors in the fitting is a subject that we have taken very seriously in our study. We have made an extensive bibliographic revision in order to decide which was the best method to estimate the goodness of the sigmoid fit. We found that numerous scientific studies from various areas use $R^2$ to evaluate the goodness of fit in sigmoid functions (e.g., Cao et al., 2019; Bhogal et al., 2014; Ritz and Spiess, 2008; Liu and Saint, 2002; Van der Graaf and Schoemaker, 1999). We have added these references (lines 293-295) in the new version of the manuscript.

Despite the generalized use of the $R^2$ method, we also analyzed its adequacy in our particular case of study in comparison with other methods. In our methodology is critical to adjust the diagonal part of the sigmoid to the thermocline and the upper straight line to the mixed layer. The lower straight line is meant to represent the deep layer, but its variations are not critical for our method to be accurate. In Figure 1e and f of the manuscript, it can be seen how the variations of the deep layer do not follow the straight line of the sigmoid, however, this is not the aim of the adjustment, and it does not compromise the validity of our results as it can be deduced from Figure 1.

We have calculated other goodness-of-fit measures (not shown in the manuscript) to validate the method, however, due to the situation described above (variations in the deep layer), we decided not to include them. The figure shown below shows the percentage of good fitting of the temperature profiles to the sigmoid function in a 2°x2° grid calculated with the Kolmogorov–Smirnov test.

[Figure]

Kolmogorov-Smirnov test (% of good fitting)

This test compares the distribution of a sample (temperature profile) with a given distribution (the sigmoid). As can be seen in the figure, this test shows a good fit in most ocean profiles (blue cells), but this test fails when the temperature profile shows variations in the deep layer, even if the method locates correctly the thermocline, as is the case with the profiles in Figure 1e and f of the manuscript. Despite this, the results obtained with both tests (Kolmogorov–Smirnov and $R^2$) are consistent and there is no test that is universally valid, as can be seen in the variety of tests used in the literature.

So in addition to $R^2$, the study provides other validations of our method: (ii) the averaged relative contribution of temperature and salinity at the most stratified point of the ocean water column (Figure 2), (ii) the comparisons of three different methodologies to locate the MLD (Figure 6), and (iii) the localization of the thermocline in temperature profiles with the proposed method and the VRI method (Figure S2 in the Supplementary information). We believe that with all this information clearly presented in our manuscript we compensate the possible limitations of $R^2$ and more importantly, we clearly show to the readers the limitations of out method.

The examples shown in Figure 1 in the manuscript are all partial profiles, which can be misleading because it is unknown to the readers whether the thermoclines are fully included. Also, in Figure 1 the thermoclines are all thin, less than 100 m. When it comes to thick thermoclines, such as shown in the figure of the supplement file, the thermocline lies between ~50 m to ~400 m, and it is possible that the depth of twice the maximum N2 doesn't cover the thermocline. The deep layer is not captured by the fitting in the supplement figure, and thus the MTD result is much shallower than the actual one. However, if the upper 500 m profile is used to perform the fitting, the upper mixed layer can not be well captured (figure not shown). The fitting depth range should be reconsidered to better present the features of temperature profiles.

We than the reviewer for pointing this out. We think that the reviewer is right as we agree that this was not clear in the manuscript. In this new version of the manuscript, we have specified that our method, by using the most stratified point of the water column, locates the strongest thermocline which in most cases will coincide with the seasonal one, and sometimes will coincide with the permanent thermocline shown in the reviewer's example (lines 3, 240, 245-247 and 319) and we believe that this precision of out method is we have corrected it.

The examples in Figure 1 are intentionally shown as partial profiles to make it easier for the reader to understand the fit of the sigmoid function to the temperature profile by using the most stratified point. These same profiles are shown up to 2000 m depth in Figure S2 of the Supplementary information, where they showed better results than the best method consulted in the background. We have now added this to the manuscript in case the reader wants to see the full profiles (line 116).

**Minor comment:**

Line 1: "… divided into three layers: the mixed layer …"

Mistake corrected (line 2).

Line 34: "plays a key role …"

In this sentence we are speaking in the plural, referring to the MLD, the MTD and the strength of the thermocline. We rewrote the sentence hoping it will be clearer (lines 33-35): "The Mixed Layer Depth (MLD), which is also the top of the thermocline; as well as the Maximum Thermocline Depth (MTD), and thermocline strength, all play a key role in…"

Line 51: "Previous regional studies have identified a shallowing and strengthening thermocline in …"

Mistake corrected (line 52).

Line 59: the meaning of MTD is not given.

Mistake corrected (lines 34).

Line 214: "…500 m in the core …"

Mistake corrected (line 216).

Line 239: "… three main layers: the mixed layer …"

Mistake corrected (lines 242).

Line 283-284: "and only gave good results with profiles located in tropical latitudes" is repeated twice.

Mistake corrected (lines 288-289).

Line 313: "… minimum and maximum depths of …"

Mistake corrected (line 319).

**Reply to Anonymous Referee #2**

Response to the reviewer comments

The Authors thank the reviewer for their comments that have helped to improve our manuscript. We hope that the reviewer finds our manuscript now suitable for publication in Ocean Science. Hereinafter, the reviewer's comments are in black, the authors' answers in blue and changes to the manuscript are shown in italics.

###########################################################################

Review of the manuscript:

"Improving the thermocline calculation over the global ocean"

Submitted to Ocean Science, September 2022

Review sent, 1st February 2023

###########################################################################

Summary:

========

This paper presents a new method to diagnose both the MLD and the thermocline characteristics (thickness, depth, strength), from observed ocean profiles of temperature and salinity. Based on Argo profiles, a new climatology of MLD and thermocline variables is constructed using this method. It is compared with previous similar climatologies of MLD to validate the approach, and then the results about the new thermocline fields (depth and thickness) are discussed.

General comments:

===============

This work is innovative because it proposes a new method that can be applied to get both the MLD and the thermocline characteristics at the global scale for the first time. This topic of ocean stratification is quite important and had several noticable studies lately (eg Sallee et al 2021), because of its potential impact on climate evolution. For those reasons, it would make sense that this rather method/data paper could be published in OS. However, when reading the paper, many serious problematic points appeared in the presentation of the method and results, showing that there is somehow a lack of knowledge about the MLD/stratification problems (unprecise method presentation, no word at all about barrier layers, comparing different types/family of MLD for validation, no introduction/references/discussion of the 3 layer hypothesis, a quite weak discussion of the results...). After having hesitated to reject the paper I must say, I would still advice a major

revision of this work because the topic is relevant and actual, even if I think that the task is big to come to a publishable version. I give below precise comments about the modifications I suggest. There are especially 4 major points (and part of those points may also be developped in the detailled comments section) that, for me, must be adressed seriously before any possible publication.

Major points to address:

====================

1) You should clarify the description of the method : (i) I have questions about the details of your method, which is the base/heart of the paper. It is very important that the method is presented clearly and univoquely, here it is not the case for me at least. There are several detailed points to address (next section of the review, fig1, eqn 1, l99-100, l143- 144) about this hereafter. Here is how I see the process of your method from a T/S profile : - get the depth D1 of vertical max of N2 from the density profile - multiply D1 by two to get the depth D2 over which we make the coming fit (D2 = 2*D1) - evaluate the direction of vertical T change --> (ii) how do you do this ? (iii) It looks not present in your python code (?) and it must be explained here, is it by computing the difference T(0)-T(D2) ? something else ? - amend the sigmoid function sign according to previous test - normalise the temperature data between 0 and 1 - the fit : non-linear adjustment of the sigmoid function f = +/- (1/(1+exp(-a*(x-b)))), to find the best a and b parameters, along with a goodness of fit coeff R^2 - get MLD by threshold on the T sigmoid (0.2 degC from 10m), get also MTD by threshold of 0.2 degC from the deepest level value of our interval (i.e. at D2 right ?) If this process is right, then what you basically do is to fit a sigmoid on the temperature profile. Then you will diagnose a temperature-MLD (or isothermal layer depth) + its thermocline, and not an isopycnal layer depth + pycnocline. .(iv) Then I do not understand why you diagnose your working depth interval (ie D2) from the density profile, logically it could be done directly from the temperature profile by searching the max of grad(T), because it is the pattern you basically look for and on which you perform a fit ? . Also by doing this, you will end up with a sigmoid that is not centered on the N2 max, i.e. your b parameter in the sigmoid fit will be different from zero, because your thermocline is rarely exactly at the depth of the N2-max (as shown in fig1f cf detailed comments). This is not what is in the text when saying l. 99 that you "locate the most stratified point in the center of the sigmoid function".

(i) Thanks to the comments of both reviewers and a reanalysis of the method, we decided to make modifications to it. Now instead of using $N^2$ to position the sigmoid and then indicating where temperature has the most control over the stratification, we use $N_T^2$ to reduce the depth range where the sigmoid will fit (lines 108-144):

*"To locate the MTD, we computed the relative contribution of temperature ($N_T^2$) to the vertical maximum of the buoyancy frequency squared ($N^2$)      to   locate   the   most stratified point from the temperature profile. We assume that this point is within the*

*thermocline, as the most stratified point of the water column given by $N^2$ is inside the pycnocline (McDougall and Baker, 2011). Schematically, most of the temperature profiles in all latitudes have a shape similar to the sigmoid function (s-shape), for this study we used the logistic function shown in Equation 1, where a is the steepness of the curve and b is the value of the midpoint of the function also known as the inflection point.*

$$f(x) = \frac{1}{1 + e^{-a(x-b)}}$$

*To perform the function adjustment, we first locate the greatest absolute value of $N_T^2$ and we take the temperature profile from the surface to its depth multiplied by two, in this way, we reduce the data from the deep layer, but making sure not to exclude the isothermal layer or the thermocline. The sigmoid function presents central symmetry with respect to its inflection point, from this point, in both directions, the sigmoid presents a diagonal line, a curve and a straight line. Given these characteristics, by fitting the sigmoid function, we seek to fully represent the mixed layer with a straight line, locate the inflection point in the center of the thermocline and consequently represent the thermocline with the diagonal line.*

*First, we evaluate the direction of the vertical temperature change. To do this, we compare the temperature value near the surface against the deeper one, if the value closest to the surface is greater, the profile decreases with depth, otherwise it increases. If the temperature decreases with depth, the sigmoid function is inverted by multiplying it by -1, then we normalize the temperature data between 0 and 1.*

*Next, nonlinear least squares is used to fit the function to obtain the optimal values of the parameters a and b. Once these parameters are obtained, it is possible to approximate the temperature values at any depth above the sigmoid. Despite the central symmetry that the sigmoid function presents, the nonlinear fit of least squares allows the fit to place one straight line shorter than the other one (without losing its shape), thus losing the symmetry and placing the inflection point in the center of the thermocline, regardless of whether or not it coincides with the greatest value of $N_T^2$. We assess the goodness of the fit with the coefficient of determination ($R^2$), this coefficient informs on how well the adjusted function approximates the real data, being 1 the best adjustment.*

*Once the sigmoid has been fitted to the temperature profile, we can determine the MLD and MTD by scrolling through the function. The temperature at a depth of 10 m resulting from the adjustment of the function is taken as a reference and is denormalized, that is, it is transformed again to be represented as a function of depth. The MLD is then determined as the depth where the potential temperature is 0.2°C higher (or lower) than the reference temperature at 10 m (de Boyer Montégut et al., 2004). To locate the MTD, we used the same procedure but going upwards in the function, in this case we take the reference temperature where the deep layer should be located and we look for the*

*difference of 0.2°C by decreasing the depth through the function. Because the method is based on a single nonlinear function adjustment, we can have a precision of even centimeters."*

This modification did not generate significant changes in the results (i.e., climatologies, comparisons between MLD calculation methods), especially since $N_T^2$ dominates over $N^2$ in most of the ocean (as already explained in the old version of the manuscript), however, there were a substantial improvement of $R^2$ where there were values less than 0.3 (e.g., Antarctic Circumpolar Current). Our answers below are based on this modification and the new results, please take it into consideration.

(ii) If a value closer to the surface is greater than the one further away, the temperature profile decreases with depth. Otherwise, the profile grows with depth. We added a detailed description of this procedure to the manuscript (lines 126-128):

*"First, we evaluate the direction of the vertical temperature change. To do this, we compare the temperature value near the surface against the deeper one, if the value closest to the surface is greater, the profile decreases with depth, otherwise it increases."*

(iii) The direction of the profile is evaluated using the relational operator greater than (>). This procedure is between lines 94-97 of the Python code (commit 1edcdc30a2fa8688d322da984414d470452803ad).

(iv) We have changed the adjustment parameter for the sigmoid. Now we use $N_T^2$ to make the adjustment from the temperature profile. We made important changes to the description of the method, as well as its step-by-step explanation, hoping to clear up uncertainties and be more clear (lines 108-144).

2) The area/conditions of validity of your method should be clarified, with possible evolution toward application on density profiles : There are detailed comments below for lines 143-152. (i) This part should be re-written I would advice, and the context of Barrier Layer should be presented as it is strongly linked to this part and it is never mentionned. Your method diagnose MLD from temperature profiles as presented here, so obviously, you will miss the density-MLD (B04 or HT09) in all BL areas, which are quite numerous despite what you say in several places of your text. But this is not a problem according to me. (ii) To get a correct density-MLD, you should simply apply your method also to the density profiles, which have also often a 3-layer shape (even more than temperature, and no density inversion occur while you may have temperature inversions). This may make a bigger paper, but as it is now I have wondered why you stick to thermocline only, trying to say that it is nearly same as pycnocline (which is not true in many areas), while it seems that you have everything to do the computation directly to get the pycnocline also. (iii) Then when you compare with previous climatologies, you should compare what is comparable, if you have a temperature-based MLD (as it is now) then you should compare with a temperature-based MLD also (eg a 0.2 degC threshold MLD compared to 10m). Otherwise you will end

up discussing Barrier Layers (BL) and compensated layers issues which have been discussed previously in papers and may not be the goal of your work (which seems to validate your new dataset of MLD and Thermocline or pycnocline if ever).

(i) We agree with the importance of the barrier layers and the relationship with our results. For this reason, we accept your suggestion and added this concept in the manuscript (lines 26-30, 42-46, 179-182, 332-346, 387-390):

"… the so-called Barrier Layer (BL) regions (Lorbacher et al., 2006). The latter, are regions where the mixed layer depth (MLD) is determined by a halocline. In these regions, the MLD based on temperature (the isothermal layer) is deeper than MLD based on density profiles (the isopycnal layer). In the opposite case, when the isothermal layer is shallower than the MLD, the vertical compensation between salinity and temperature causes compensated layers (CL) located below the MLD (de Boyer Montégut et al., 2004).

"Other classifications of the thermocline have been proposed from a machine learning approach. For instance Jiang et al. (2017) classify the thermocline due to its form as positive, inverse and mixed thermoclines as well as multi-thermoclines. The forms that originate this classification could be related to the temperature inversions that occur at the base of the BL and in the polar regions (de Boyer Montégut et al., 2004; Dong et al., 2008) and by the double-diffusive staircase stratification events (Timmermans et al., 2008; Toole et al., 2011)."

"The temperature profiles shown in Figure 1a-e were taken from high latitudes to near the equator, while the profiles in Figure 1f-j are located in the North Pacific region during winter months, where thick BL are found (see de Boyer Montégut et al. (2007))."

"… There, the method of B04D strongly differed from other ones, we attribute this to the fact that staircase stratification have been reported in these areas (Timmermans et al. 2008; Toole et al., 2011), defined as BL and CL regions (de Boyer Montégut et al., 2004). Indeed, our preliminary results have shown inverse thermoclines, multi-thermoclines and mix thermoclines (Jiang et al., 2017) in these regions (not shown). In this sense, we have provided some exampled of how our method could accurately locate the BL in some cases (Figure S1 of the Supplementary information). Tin these cases, the MLD calculated by the 0.2°C threshold (B04T) coincides with that calculated with our methodology. Conversely, in other cases, where the isothermal layer was highly variable (Figure S1g and i) our method was unable to locate the therocline. The BL was also located to these profiles using the $D_\sigma$ and $D_{T-02}$ criteria defined in (de Boyer Montégut et al., 2007), showing that when the thickness of the BL is of the order of hundreds of meters, the proposed method locates the inverse thermocline produced by the inversion of temperature that is found below the isothermal layer. On the contrary, in thinner BL, our method locates the thermocline below the lower limit of the BL. Sprintall and Tomczak (1992) define the BL as the distance that separates the

*MLD from the MTD, however, when this BL is thick, the proposed methodology locates the inverse thermoclines that are within this barrier and not those that could be below $D_{T-02}$"*

*"The formation of intermediate strength thermoclines (i.e. 0.2 < TSI < 0.8) in the North Pacific and North Atlantic coincides mainly with the months (July-September) when there are no BL in these regions (de Boyer Montégut et al., 2007). On the contrary, from January-March when the BL are thicker (de Boyer Montégut et al., 2007), weak thermoclines and regions with inverse thermoclines are shown."*

(ii) We agree that the density profiles can also be represented in three layers as mentioned in lines 21-22, and that the temperature profiles may contain temperature inversions, however, in the manuscript this topic is treated as inverse thermoclines. For a better understanding of the relationship between these two concepts, we added the concept of temperature inversions and a broader explanation of thermocline classification to the manuscript (lines 42-46, 332-346, 385-396).

Finally, although it seems that the density profile would better fit the sigmoid function, this is not the case. Another function is needed that makes a better adjustment, since in tests carried out on the density profiles they did not give good results, not even locating the MLD. This is not shown in the manuscript, since the aim of the manuscript and the method is the thermocline. We made changes in this version of the manuscript and the method, so that the method does not depend on the density profile for its fit (lines 108-144).

(iii) We agree that comparison of our MLD results with a temperature-only method is necessary. We will expand the answer of this point together with the next one below.

3) (i) As stated above, you should compare climatologies with at least same family of criterion : Your criterion is a threshold criterion of 0.2 degC on a fitted temperature profile. You should then compare it to the 0.2 degC threshold criterion of B04 also on temperature profiles not to the 0.03 kg/m3 density criterion, or you will mix several source of differences, especially salinity effects (cf lines 175-182). B04 and HT09 are shallower than your temperature-MLD mostly in the BL areas and high latitudes, which makes sense. B04 is deeper than HT09 which also makes sense as HT09 is a mix of several criteria and takes the smaller one, and this has also been already noticed in HT09 paper. (ii) Here you want to validate your MLD method, not discuss where salinity plays or not, for me it is another problem that has been tackled in many Barrier layer papers already. So you should compare your temperature based MLD with a similar temperature based MLD (eg deboyer 2004 0.2 degC threshold). (iii) Then if you want to have a part to see where your temperature MLD is biased compared to a density MLD, you should first cite the BL papers, and then make a comparison that should well correspond to BL areas as is seen in your figures already.

(i) We now added the $0.2°C$ criterion to the manuscript comparisons such as B04T (lines 149-161, 190-203, 229-241, 245-248, 256-263, 324-327, 338-339, Figure 5 and S1, S4 and S5). However, the B04D (before B04) method is currently the most widely used method to

locate the MLD and for this reason we believe that it should also be present in the comparison, if the results of our method deviate too much from those of B04D it would be worrying (except for BL regions of course). The results of HT09 may or may not come from the temperature profile, depending on the criteria of the method, so it is not so far from our method. In fact, HT09 results are more related to those of our method and of B04T than to those of B04D. (Figure S5). We think that adding this fourth method to the comparison seems to give robustness to the validation of the MLD calculation. However, this does not represent major changes in the results, since our method locates the temperature inversions as inverse thermoclines, so the BL does not sink our MLD calculations (Figure 5 and S1) and for that reason it is so highly correlated with methods B04D and HT09 (Figure S5).

(ii) We agree with you, for this reason we made modifications in the manuscript due to the change from $N^2$ to $N_T^2$ (lines 108-113) and we removed the sections where we discussed whether salinity plays or not (lines 165-168, 189-200).

(iii) Suggestion accepted (lines 26-30, 42-46, 179-182, 332-346, 387-390).

4) An extended discussion about what you expect to diagnose as your thermocline is necessary, along with presenting in introduction what you talk about with this 3 layer hypothesis. Will you give the permanent thermocline or the seasonal one ? is it just below the MLD or not ? is it a mix of those 2 features ? (see helber et al JGR 2012, sprintall cronin EOS2001, Johnston Rudnick JPO 2009...)

In this new version of the manuscript, we have specified that our method locates the thermocline that is just below the MLD and using the most stratified point of the temperature profile, locates the strongest thermocline that in most cases will coincide with the seasonal one. In addition to locating inverse thermoclines where temperature inversions are found (lines 309-310, 342-343):

*"In this study we have proposed a new method to locate the strongest thermocline that lies just below the MLD of the water column."*

*"... the proposed method locates the inverse thermocline produced by the inversion of temperature that is found below the isothermal layer."*

Other detailed points to address :

============================

l. 22-23 : the ocean surface layer density and extent of the MLD is also determined by salinity in all "Barrier Layers" area which are present all over the tropical ocean (e.g. Sprintall and Tomczak JGR 1992, de Boyer Montegut et al JGR 2007). This should be mentionned here, as the total areas where salinity drives the stratification and MLD is actually not small at all

(maybe about 20% of the total ocean surface when checking maps visually in the 2 references above).

Suggestion accepted (lines 26-30).

l. 26 : "The characteristics and … (coast-ocean gradients)" --> I think this sentence does not bring much information actually but could be kept as a transition. However, I would have said "spatially" instead of "latitudinally and longitudinally". Also, I do not really get the meaning of the term "coastocean gradients" here. It may be worth explaining further or being more precise, or else remove it.

This sentence was written to note spatial variations. Suggestion accepted (line 34).

l. 29 : I advice the change: "heat flux" --> "net heat flux"

Suggestion accepted (line 38).

l. 59 : "MTD" : I could not find the definition for this before (guess it is for "Maximum Thermocline Depth")

Mistake corrected (line 47).

l. 62 : I am surprised that I could not find any trace of the Jiang et al 2016 publication via google, may be my mistake, but talking about maximum curvature method, I also though of the paper by Lorbacher et al JGR 2006, and would have liked to see how the Jiang paper would compare to that one, with the possibility of citing also the Lorbacher paper as being before in time, if it suits the situation of course.

The maximum curvature point method is used in different areas of knowledge and for different purposes. In this case (lines 75-79) we cite the study by Jiang (2016) because he used this methodology to locate the MTD in a region, on the contrary, Lorbacher et al. (2006) used this methodology for demarking the MLD. We include his study as part of the background to this calculation (lines 73).

l. 79 : you could also tell what quality control flag you use for selecting the acceptable DMQC data, only the flag 1 ? both flag 1 and 2 ? (of course you removed the 3 and 4 QC flags data I suppose)

Since we recalculated all the values of MLD and MTD with the change to $N_T^2$, we decided to change all the data that were not coming from flags 1 or 2 to NaNs to give more reliability to these results using only good and probably good data. This is now indicated on lines 94-95, 173-174:

*"… we used the profiles already evaluated by the delayed mode quality control (DMQC) from January 1998 to December 2021 (more than two million), that have been classified as good or probably good data."*

*"This methodology developed here was applied to each of the DMQC Argo profiles marked as good or probably good data."*

Figure 1 (and equation (1) of sigmoid): - from the description of your algorithm, you multiply the depth of density stratification maximum by 2 to get the max depth on which you fit you sigmoid, but on this figure, you do not show profile on this complete depth in (a) to (d) panels (e and f look quite ok to me). For exemple, in panel (a), the max stratif N2 (center of sigmoid) appears to be at 80 meters depth, so we could have expected that you show the profile on the whole depth of the sigmoid fit (ie 160m, but you stop your plot around 140m), and same in b,c, and d. Is there a reason for that ? if not, I think it could be somehow better to see the whole depth on which the fit occurs.

We have updated Figure 1 and the description of the method, hoping to clarify these inconsistencies (lines 108-144).

The examples in Figure 1 are intentionally shown as partial profiles to make it easier for the reader to understand the fit of the sigmoid function to the temperature profile. These same profiles are shown up to 2000 m depth in Figure S1 of the Supplementary information. We have now added this to the manuscript in case the reader wants to see the full profiles (line 145):

*"To visualize the profiles of Figure 1 up to 2 000 m depth, see Figure S1 of the Supplementary information."*

- (i) in panels b and f at least, it looks like the max of N2 is not the center of the sigmoid, with the limits of the thermocline you diagnose being even below this depth fro panel f. This is apparently in contradiction with the algorithm you describe, or I may have missed something (as said in your texte you multiply by two the N2 max depth so that your sigmoid is centered at the max of N2, no ?). This may show the difficulty of mixing a density diagnostic (max of N2) with a temperature one (fit on T profile) because the 2 are not always linked, and their maximum gradient can often be at a different depth. This may also be linked to the fact that your actual sigmoid is not necessarily symetric with the max of N2, which is necessary because you fit on T-profile, not on density-profile. (ii) Indeed, if I believe the formula found in the python code of Romero 2022, it is "1/(1+exp(-a*(x-b)))", so you should give this formula in your equation (1) so that readers know exactly what variables you fit (ie a and b here), and that your final sigmoid is not necessarily centered on the max of N2, but centered on the max of grad(T) as you make the fit on T, not on density.

(i) Thanks for pointing this out as this is caused by a not very detailed description of the method. We have modified the method description hoping to be clearer (lines 108-144).

(ii) In the old version of the manuscript the general function of the sigmoid was presented. The function contained in the Python code is the logistic function, one of many sigmoid functions. We accept your suggestion and specify the function we use (lines 113-116,

Equation 1). We also added a more detailed description of the fit of the function by nonlinear least squares (lines 130-134).

- as an illustration, I would advice to also show a typical exemple of a profile when your method does not work so well as it happens quite in several extended places, i.e. when the 3-layer shape is not valid for temperature (R2 below 0.7 or 0.5), this should happen a lot in high latitudes beta-ocean with many interleaving features and several temperature inversions, or maybe in cases of remnant layers when restratification occurs by steps during spring at mid latitudes.

Suggestion accepted, we added an example in Figure 1g and it is discussed in lines 184-203:

*"Profiles in Figure 1f and g show the greatest variability in $N_T^2$, but the quality of our adjustment differs between them. In panel f, despite the high variability in the deep layer, the methodology correctly determines the MLD and MTD. However, in panel g, high variability occurs from the end of the isothermal layer, and our methodology cannot perform the adjustment of the function correctly. In the same way, the B04T, $D_\sigma$ and VRI methods failed to correctly locate the MLD and the thermocline (as shown in Figure S1)."*

l. 119 : "HT03" should be "HT09" I think

Mistake corrected (line 152-153).

l. 120 : here it would be better to (re)precise also the reference depth of B04 as the scope of the paper is about methods to diagnose MLD and stratifications, e.g. "... a threshold of 0.03 kg.m-3 compared to the reference value at 10 m depth in the density..."

Suggestion accepted (line 155).

l. 121 : "...de MLD..." should be "...the MLD..."

Mistake corrected (line 157).

l. 136-137 : from your figure legend lon/lat data, I have profiles a, b and f in southern hemisphere, and d and e in northern, so you should correct something somewhere to be sure where we are.

Mistake corrected (lines 179-182).

l. 140-141 : Your conclusion is somehow a scientific overstatement. You cannot draw a general conclusion from a sample of 6 profiles among 2 millions or more. This analysis is not scientifically reproductible. If you take another 6 profiles, you may have a completely different conclusion. So it cannot be used for a general conclusion. Such exemples of profiles are nevertheless important to be shown so that the reader can realize some of the main details of the method illustrated in different situations (for exemple the fact that in austral the max of N2 is not at same depth as the max of thermocline, cf remark above for figure 1 also). I would advice that you temper your conclusion saying that this figure is an

illustration of how the method works, and of the fact that it seems to work well (but this will be quantified afterward in the paper actually).

Suggestion accepted (lines 203-216):

*"To illustrate the precision of our method and to identify regions where it should be applied with caution due to the variability of the temperature profiles, we provide a map of the monthly average of $R^2$ (Figure 2).*

*In general terms, the adjustment of the sigmoid function is very good (with $R^2 \geq 0.9\$$) in low and mid latitudes. However, the cells with red and gray colors should be taken with caution. These present $R^2 < 0.3$ and $< 0.7$ respectively, which indicates that the adjustment of the sigmoid function was poor or not optimal. The worst adjustments correspond to the core of the Antarctic Circumpolar Current in the Southern Ocean, the North Pacific and the Western North Atlantic. These are regions where the stratification of the water column is dominated by salinity, there are temperature inversions and/or present strong currents and associated turbulent dynamics. In general terms, in the regions where the adjustment was worse, it was less good in winter months.*

*Due to the results of the preliminary evaluation, which showed that both the visual examination (not shown) and the values of $R^2$ indicate that our methodology correctly locates the MTD and the MLD at different latitudes. After this first step, we carried out the validation against other methodologies. "*

His conclusion is a bit of a scientific exaggeration. A general conclusion cannot be drawn from a sample of 6 profiles out of 2 million or more. This analysis is not scientifically reproducible. If you take another 6 profiles, you may have a completely different conclusion. Therefore, it cannot be used for a general conclusion. However, it is important to show these sample profiles so that the reader can see some of the main details of the illustrated method in different situations (for example, the fact that in austral the N2 maximum is not at the same depth as the thermocline maximum, cf previous comment for figure 1 as well). I advise you to temper his conclusion by saying that this number is an illustration of how the method works and the fact that it seems to work well (but this will be quantified later in the article).

Our intention here was to describe the results of the preliminary evaluation and validation of the method, not to draw a general conclusion. That is why the calculation of the MLD on a global scale is validated below and later the results of the comparison of results between our method and VRI are discussed. We rewrite this paragraph hoping to be clearer (203-216).

l. 143-144 : Again, for me this statement is not exact or misleading (and same at l.99-100). From fig 1. and formula in your python code, we can see that your sigmoid is not necessarily symetric with the max of N2 being the central point of the sigmoid (not central if b param

is not zero in your sigmoid, cf also major point to address). Eventually I do not see exactly why you need to show that your method should be applied with caution in salinity dominated region. This is basically obvious from your method : you diagnose the isothermal layer depth (or temperature-MLD) because you work on the Tprofile only, so of course in every Barrier Layer areas, you should end up with a too deep temperature-MLD compared to a density-MLD, and this may happen in all high latitude areas (beta oceans) and all tropical and mid latitude Barrier Layers areas (cf maps by sprintall tomczak JGR 1992 or deBoyer Montegut et al JGR 2007). Eventually your method is another way to compute the MLD (here based on temperature, but it could/should have been applied on density also) with one advantage that is to give also an estimate of thermocline depth and thickness which is a nice plus of your method of course.

We agree with your suggestions. We changed the description of the method and rewrote the discussions about regions dominated by salinity (lines 108-144, 165-168, 189-200).

Figure 2 and l. 144-152 : Here you show the T/S contributions to the stratification at the depth of the max-N2 (max density stratification). If we assume that this depth is most of the time at the base of the isopycnal mixed layer, then this analysis is very similar to the Barrier Layer (BL) mapsmentionned above. It is then surprising that all the historical BL in western equatorial paicifc are not more pronounced, but still they are present in your plot and should be noticed (As your map is not seasonal, only the permanent BL patterns are well evidenced, or as the diagnostic is not really the same there are small differences, but in fig 8 the shaded areas indeed occur in west equat pacif during jun to sept for exemple.)

This figure and discussions were removed (lines 165-168, 189-200).

l. 151-152 : This statement is misleading to me. Your present diagnostic does not tell you about the vertical thermal structure of the ocean. You may have a BL situation typically in the west equat pacif, where you have warm waters at the surface then a thermocline with decreasing temperature at about 60m and colder water below (a thermal classic three layer structure). Yet you also have fresh water eg in the top 20 or 30 meters, with a 1st pycnocline at this depth, dominated by salinity. What figure 2 tells us is just what happens at the depth of N2 max. This is rather the R2 that tells you if your T-profile is well fitted to a sigmoid and hence if it has a 3-layer shape. Eventually I wonder about the relevance and this figure in your work, or it needs to be better presented, within the context of BL I would say, and may be seasonally.

Suggestion accepted (lines 189-200).

l. 193-195 : I do not really agree with this statement, and this is not what was shown in HT09 and from your map in fig S1c, where B04 density MLD is shown to over estimate the MLD in high latitudes. This also makes sense with the fact that the ocean has a very low vertical stratification at high latitude, which pushes for smaller density criterion to get the mixed layer, especially in winter. And even if Perralta-Ferriz and Woodgate 2015 used 0.1 kg/m3

(but based on a heuristic method for choice), they recognize that they had to use a smaller 0.03 kg/m3 for several of their winter profiles to avoid over estimation of MLD. Also you show that B04 overestimate MLD in arctic ocean in winter (region 48 of fig 6 a b), and then at line193 you say that their 0.03 criterion underestimate the MLD in polar regions. This makes not sense to me.

*Figure S1 was changed by seasonal relative differences between our method and HT09 (Figure S2), B04D (Figure S3) and B04T (Figure S4), and a correlation matrix was added between the four methodologies (Figure S5). Underestimations, overestimations, and correlations are now discussed taking into account Figure 5 and S2-S5 (lines 229-263):*

*"In general terms, our climatology agrees with those of Holte et al. (2017), de Boyer Montégut et al. (2004), HT09, B04D and B04T (Figure S2-S5 in the Supplementary information), but there are some differences. The comparison between our method and HT09 (Figure S2) shows no net underestimation or overestimation and the relative difference between the two methods is less than |25%| over most of the ocean. The HT09 method gives slightly shallower mixed layers than our method in subpolar latitudes and deeper in tropical and subtropical regions. On the other hand, B04D and B04T generally overestimate the MLD with respect to our method (Figures S2 and S3). B04D has the greatest differences (above 50%) compared to our method, especially in the northern hemisphere from July to September, while B04T presents its greatest differences in tropical and subtropical latitudes throughout the year. To make a more quantitative comparison, the MLDs computed with the four different methods were averaged within the reference regions of AR6-WGI, as these were designed for regional synthesis (Figure 4).*

*The red-delimited regions are fully continental regions and were not used for our analysis. The regions with less than 10 averaged values were also excluded. The averages of the MLD computed with each method in each region were plotted for a representative month of each season (Figure 5).*

*The MLD shows good agreement between the four methods in most regions. In general terms, the method proposed here is in better agreement with HT09 and B04T, being B04D, the one that exhibits the greatest differences. In February and May, the B04D method seems to overestimate the MLD in the regions of Northeast North-America and the Arctic-Ocean (regions 03 and 48, Figure 5a and d). These are polar regions containing semi enclosed seas in the Northern Hemisphere. It is likely that these regions exhibit particular dynamics that complicate the detection of the MLD with a global threshold based method. One possible explanation is that coastal regions generally are worse sampled, but also, that the computation of the MLD can be complicated by a number of coastal processes such as river discharges or shallow bathymetry among others. Moreover, the traditional delta density criterion of $0.03\ kg\ m^{-3}$ has been suggested to underestimate the MLD in polar regions, as demonstrated in the study by Peralta-Ferriz and Woodgate (2015), where it was found that a better criterion for these regions is $0.1\ kg\ m^{-3}$. Interestingly, the agreement*

*between the four methods is also good in the regions where our adjustment was not considered to be good ($R^2 < 0.7$, Figure 2). In other complicated region as the one around Greenland (region 01) there are some differences between the four methods, but the one proposed here either agrees with B04T in February and HT09 in August, and gives an intermediate MLD value with respect to two other methods (as in May and November). The Spearman correlation between the results was calculated (Figure S5), and showed high correlation between the results of all the methodologies, with the results of HT09 and B04T being the most correlated (0.98), followed by the proposed method and B04T (0.95). Finally, all the correlations between B04D and the rest of the methodologies showed values close to 0.90."*

Figure 6 : (i) I advice to replace this one and fig S1 by a plot of the maps of differences between HT09/B04 and your method, at 2 relevant seasons at least, which makes 4 maps, and enables a more detailed analysis than the (ii) AR6-WGI regions (why choosing those?), that were not designed for MLD studies and are not really relevant for that (for exemple dividing the labrador sea deep convection region in 2 parts)

(i) We accept the suggestion to modify Figure S1. Figure S1 was changed by seasonal relative differences between our method and HT09 (Figure S2), B04D (Figure S3) and B04T (Figure S4). The similarities between HT09/B04D/B04T can be seen in Figure 5 and S5.

(ii) This set of reference climatic regions has been used since they were designed for regional synthesis. In addition, places such as the Labrador Sea (region 01 and 03), the Greenland Sea (region 01) and the Arctic Ocean (region 48), do not contain too many profiles to divide them into subregions. We believe that using these reference regions, we are considering both the regions of relevance to the MLD (that you mention), as well as other less relevant regions, in order to validate the method. For a more specific comparison, Figures S2-S4 are available.

l. 204-205 : As a remark here, I would say again that your MLD method validation is not the main point fo your article. Basicaly, your method for MLD diagnostic is the 0.2 degC threshold on a temperature profiles (after a sigmoid fit, that is your plus so as to diagnose thermocline after), so it is nothing really new for me, and it should be validated with a previous climatology of the same type (a threshold of 0.2 degC from T profiles), and should work with no big surprise to me. The important and new part of your paper is the thermocline (or pycnocline if you add it) part that comes now indeed.

We agree with you. We added the $0.2°C$ method under your suggestion in the comparisons. However, in the absence of a robust and comprehensive method to compare our MTD estimates, validating the MLD results is one way to increase the reliability of our method.

Figure 7 : In may an extended pattern of MTD reaches 350 to 400 m in north atlantic, while it is about 100m or less in MLD (linked to the about 300 m thickness in figure 8 also). At that time restratification already started form the MLD shift from April to May, then I wonder

why you do not catch this restratification with your method. I would have expect to see the MTD not so far from the MLD in may when the restartification signal is there in MLD, why is it not the case ? Are you diagnosing the stratification below the MLD (which I doubt actually) or a stratification that corresponds to the permanent thermocline ? You must talk about this somewhere, and also you should define your view of the three layer more precisely, because this view corresponds to warm surface / permanent thermocline / deep cold ocean, but with your method that limits the fit to max of N2 depth * 2 we would think that you may not catch the permanent thermocline but the seasonal one (or trnasition layer after Johnston and rudnick JPO 2009, see also Helber et al JGR 2012 that tackled also a related topic). For such a discussion you can also rely eg on Sprintall & Cronin EOS 2001 (upper ocean vertical structure) for a basis.

Suggestion accepted (lines 309-310, 342-343).

l. 213-218 : this description does not really bring something new to the paper, and explanations are quite well known already

That's right. But the paper shows a new methodology to calculate the MLD and MTD. If the results obtained agree with current knowledge, it is an indicator that the methodology correctly calculates these depths, otherwise, the method should be reviewed. This paragraph was written to explain to the reader why these MLD and MTD are computed in the winter months in those places.

l. 220-221 : same remark as above, here if you have very close values of MLD and MTD (either small or even large), then it means the difference MLD-MTD is small and based on your sigmoid that is symetric, 2*diff (which is your thickness) is also small, nothing special that is worth noting for me here, or I missed something.

Suggestion accepted (lines 286-287).

l. 239 : your term "mixing" here is not correct. the mixed layer is something different than the mixing layer and you should be aware of that when writing a MLD/thermocline paper I suppose. See Brainerd and Gregg DSR 1995 "mixed and mixing layer depth" to know about this point and make the difference between the 2.

This was a writing error, here we refer to the mixed layer. Mistake corrected (line 311-312).

l. 259-266 : (i) precise RTQC and DMQC i do not think it has been done before. (ii) Also I doubt when you say "a large number of cells would contain less than 3 values for monthly averages". I do not have the same result on my side, I have a map with number of argo profiles per 1 degree boxes annually and it is over 50 over nearly the whole ocean, which means on average over 4 profiles per month at least. So I have difficulties to understand from where you make this statement.

(i) The meaning of DMQC is in line 94. Now we added the meaning of RTQC in this paragraph (line 348).

(ii) Below we show a map of the number of DMQC profiles in a 1°x1° monthly grid.

[Figure]

The figure shows in reddish tones, the cells that contain less than three profiles in each month. The cells in which at least 1 profile has been measured were detected, so as not to take into account cells where the Argo floats are not capable of measuring. Of the total of 409 104 cells in a 1°x1°x12-month grid, 132 425 cells have less than three profiles, that is 32% of the cells. We rewrote this sentence, hoping to be clearer (line 349-351):

*"This smaller-size grid is not optimal to be used with the amount of DMQC data available in this snapshot (Argo 2022b), as about a third of cells would contain less than three values for monthly averages."*

To reproduce our results, please be sure to use the same snapshot that we quote in the manuscript (Argo, 2022), otherwise this could lead to differences in the results.

l. 267-269: Your argument is not correct here. Oceanographers have pushed for salinity measurements historically because the latter also plays a role eg in MLD and now that we

have those data (with argo), you say "it is better to not have it and just use temperature" and for that you rely on a very good but very old paper (rao 1989) at a time, especially in indian ocean, when they had no choice at all about using temperature and/or salinity for diagnosing MLD. This has nothing related with a scientific choice. Also if you rely on temperature only you must face the BL problem that is more extended than compensated layers, and about which you must talk somewhere.

Suggestion accepted (lines 356-360):

*"Using the density to estimate the MLD usually gives good results, since it depends on temperature and salinity, however, the density can show vertical compensation below the well-mixed layer (de Boyer Montégut et al., 2004) causing deeper MLDs calculated from density thresholds. Although B04D and HT09 methods use density profiles for the calculation of the MLD, this does not give good results with the methodology proposed in this paper."*

L. 271-272 : This is not convincing me. For me it is the temperature profile that is more complicate to fit with a sigmoid, as it can have many features (interleaving in high latitude, several inversions, fossil/remnant layers etc...), while the denisty profile do have a simple chape as the ocean is stable, so light waters are always above heavy ones, and the density profile is the one that have by default the shape of the sigmoid. If it is not the case for you, then you need to show precisely why this is not true and where. As I said before, I would expect that you do your method on density profiles also, so I really do not understand why it cannot work for those. You must discuss about this with clear evidences of the fact that fitting T is easier than fitting density.

Suggestion accepted, we add a broader explanation of why this method cannot be applied to density profiles (lines 360-370):

*"The adjustment of the sigmoid function bases on the typical shape of the temperature profile. In order to calculate the pycnocline with a similar methodology to the one proposed here, it would be necessary to find another function that better fits the density profile, since despite also being able to be represented in three layers, the typical density profiles present an inclination along the entire profile, which makes it difficult to fit a conventional sigmoid function."*

l. 282 : see Helber et al JGR2012, they study the startification below the MLD and they have fields of stratification below the MLD as I remember.

Helber et al. (2012) defines the thermocline as the transition layer (paragraph 11). However, later (paragraph 31), they mention that the TLT (transition layer thickness) used in their study may encompass the entire thickness of the thermocline. So, we could not make a direct comparison between our thermocline thickness and TLT, as we did with the MLD calculation methodologies. However, we believe that the TLT climatology (Figure 16 in their

study) presents great similarities with our climatology of the thermocline thickness (Figure 7), so we add it in the discussion (lines 376-381):

*"Helber et al. (2012) mention that the transition layer thickness (TLT) used in their study may encompass the entire thickness of the thermocline, and in fact their TLT climatology presents some coincidences with our climatology of the thermocline thickness (Figure 5}). The most notorious coincidences are in the regions of the northeast Pacific Ocean, the northern North Atlantic Ocean and the Antarctic Circumpolar Current in the Southern Ocean, where the thermocline thickness and the TLT show values greater than 350 m in the winter months, in addition to the marked seasonality that both presents in tropical and subtropical regions."*

l. 283-284 : repetition of sentence, problem of edition to correct

Mistake corrected (lines 391-392).

Conclusion : it should be extended, especially talking more about the real new point of your article which, again, is not the MLd for me, but the thermocline (and hopefully pycnocline) characteristics that you estimate here.

To finish, you never show a map of the thermocline strength, while you diagnose it (as slope at the center of the sigmoid for exemple), why not ? I think this is a new interesting field/variable that must also be shown and discussed here as it gives the strength of the link between the ocean surface and interior, it would be a plus of your paper.

Thank you for bringing this point to our attention. We included the monthly climatology of the thermocline strength in the manuscript (Figure 8, lines 168-171, 296-307, 381-390). This climatology was calculated with the Thermocline Strength Index (TSI), we did not use the slope of the sigmoid diagonal because our criteria to locate the MLD and MTD through the sigmoid is a threshold of 0.2°C. Therefore, the slope between these two depths and the diagonal do not coincide:

*"Once the calculation of the MLD was validated, the monthly climatologies of the MTD, the thickness and the strength of the thermocline were obtained. The thermocline strength was calculated using the thermocline strength index (TSI), defined as $\triangle T/\triangle h$, where $\triangle T$ and $\triangle h$ are the differences in temperature and depth, between the MLD and MTD (Yu et al., 2010)."*

*"Finally, the climatology of the thermocline strength (Figure 8) maintains the seasonality in subtropical and subpolar latitudes seen previously with a |TSI < 0.1|. On the other hand, the Tropical Eastern Pacific and Tropical Eastern Atlantic have TSI > 0.1 throughout the year, as do the North Pacific and North Atlantic, but from June-November. The Black Sea shows the highest values of TSI (> 1), while the lowest values (< -0.1) are scattered in subpolar regions and in the discharge region of the Ganges-Brahmaputra rivers (December-February)."*

*"Regarding the climatology of the thermocline strength (Figure 8), it was calculated through the TSI, this index indicates the steepness of the thermocline (Duka et al., 2021). The further*

*TSI is from 0, the slope is less steep and therefore the thermocline is stronger. The strongest thermoclines found in the Black Sea are associated with thin thermoclines (15-20 m) between warm surface waters and cold intermediate waters (20-8°) (Akpinar et al., 2017) which produce small slopes. Negative values of TSI are caused by inverse thermoclines, these were found mainly in subpolar regions and in the Ganges-Brahmaputra rivers discharge, these regions present TSI close to 0, which means steep slopes and therefore weak thermoclines. The formation of intermediate strength thermoclines (i.e. 0.2 < TSI < 0.8) in the North Pacific and North Atlantic coincides mainly with the months (July-September) when there are no BL in these regions (de Boyer Montéggut et al., 2007). On the contrary, from January-March when the BL are thicker (de Boyer Montéggut et al., 2007), weak thermoclines and regions with inverse thermoclines are shown.*"

##############################################################################

Refences cited in this document:

Argo: Argo float data and metadata from Global Data Assembly Centre (Argo GDAC), 2022.

Jiang, B., Wu, X., and Ding, J.: Comparison of the calculation methods of the thermocline depth of the South China Sea, Mar. Sci. Bull, 35, 64–73, 2016.

Lorbacher, K., Dommenget, D., Niiler, P. P., and Köhl, A.: Ocean mixed layer depth: A subsurface proxy of ocean-atmosphere variability, Journal of Geophysical Research: Oceans, 111, 2006.

Wong Annie, Keeley Robert, Carval Thierry, Argo Data Management Team (2022). Argo Quality Control Manual for CTD and Trajectory Data.

---

## Referee Report (RR1)

##################################################################################

Review of the manuscript "Revised submission":
"Improving the thermocline calculation over the global ocean"

Submitted for publication in "Ocean Science"
2[nd] Review, 19th April 2023

##################################################################################

After revision of the paper it appears to me that this work is more solid and precise when presenting its new methodology and approaches. It also encompasses nice discussions about the role of salinity on MLD estimations, which are necessary in a context of diagnosing MLD from temperature profiles. Main issues that were raised in the first review were correctly adressed and the reviewer thanks the authors for their detailed answers. This work is then quite ready to be published in Ocean Science according to me.

I have still a few comments below and also suggestions for minor/technical corrections, especially about figure 1 and S1 that, for me, could be better described and be coherent with each others :

l.6 : instead of "the maximum depth of the thermocline" what you compute more exactly is "the maximum thermocline depth" or identically "the depth of the maximum thermocline"

l. 102-103: I find the formulation here a bit unclear. I would suggest instead something like : "To locate the MTD, we computed the vertical maximum of the contribution of temperature to the squared brunt-Vaisala frequency (i.e. maximum of $N_T^2$) to locate the most stratified point from the temperature profile."

Also, it is quite obvious what is $N_T^2$ but still there is no explicit routine to compute it in TEOS10 as far as I have checked, so it could be nice to recall the formulation of $N_T^2 = - g^2 * rho * alpha * d(Cons.Temp.)/d(press)$.

l.214-215 : this sentence is not clear grammatically I think, and should be rephrased

l.338 : "Tin" → "In"
l.340 : "thermocline"

Figure 1 : when comparing figure 1 with figure S1 (which is a plus to have it), I do not find the two corresponding upper/lower thermocline limits sometimes and also the colour code is not the same it seems (upper/lower is resp. black/red in fig1 and in figS1, it looks to be the opposite, so you should arrange this to be same). For exemple, on fig S1b, there is only one black dotted line which seems to be the one around 40m but the 400m one is not there, why ? better to have it in both figures , same for figS1c (missing red line) ; same for fig1g which has only the black line while figS1g has also one red one close to surface ; same for figS1f which has very different lines that on fig1f ….

Also the legend of fig1 and figS1 should describe further what are those lines. You write about thermocline and mld or barrier limits on fig S1 but you never say to what they correspond on the figure. This should be corrected as for now it is unclear and misleading to me.

Figure 8 : missing unit of the variable above the colorbar, can be e.g. "Thermocline Strength Index (TSI) [degC/m]"

##################################################################################

---

## Author Response (AR2)

**Reply to Anonymous Referee #1**

Response to the reviewer comments

The Authors thank the reviewer for their comments that have helped to improve our manuscript. We hope that the reviewer finds our manuscript now suitable for publication in Ocean Science. Hereinafter, the reviewer's comments are in black and the authors' answers in blue.

Thank the authors for the detailed reply to my comments, the reply and revision give convincing answers, and I have no further major comments.

**Minor comments:**

Line 6: typo, should be 'strength'.

Mistake corrected (line 7).

Line 85: The structure or content of the article should be introduced at the end of the Introduction section.

Suggestion accepted (lines 86-88).

*"In this study, we first describe the proposed method to calculate the MLD and MTD, then we compare the results with other methods found in the literature, finally we calculate the thickness and strength of the thermocline, to obtain the climatologies of the mixed layer depth, the maximum thermocline depth, the thermocline thickness, and the thermocline strength index."*

Line 250-251: The sentence 'Finally, the climatology … |TSI<0.1|' is confusing, what does 'seen previously' refer to?

We removed the phrase "seen previously", hoping to be clearer in this sentence (lines 261-262).

*"Finally, the climatology of the thermocline strength (Fig. 8) maintains the seasonality in subtropical and subpolar latitudes with a |TSI < 0.1|."*

Line 280: 'exampled' should be examples.

Mistake corrected (line 291).

Line 281: typo, 'Tin' should be 'In'.

Mistake corrected (line 292).

Line 283: typo, 'therocline' should be 'thermocline'.

Mistake corrected (line 295).

Figure 1: The location of figures 1(e) and (g) are in the equatorial and subtropical regions, respectively, but their surface temperatures are low, please clarify it.

The order of the coordinates in the caption was wrong. Mistake corrected (Fig. 1 and lines 170-172).

**Reply to Anonymous Referee #2**

Response to the reviewer comments

The Authors thank the reviewer for their comments that have helped to improve our manuscript. We hope that the reviewer finds our manuscript now suitable for publication in Ocean Science. Hereinafter, the reviewer's comments are in black, the authors' answers in blue and changes to the manuscript are shown in italics.

##############################################################################

Review of the manuscript "Revised submission":

"Improving the thermocline calculation over the global ocean"

Submitted for publication in "Ocean Science"

2nd Review, 19th April 2023

##############################################################################

After revision of the paper it appears to me that this work is more solid and precise when presenting its new methodology and approaches. It also encompasses nice discussions about the role of salinity on MLD estimations, which are necessary in a context of diagnosing MLD from temperature profiles. Main issues that were raised in the first review were correctly adressed and the reviewer thanks the authors for their detailled answers. This work is then quite ready to be published in Ocean Science according to me.

I have still a few comments below and also suggestions for minor/technical corrections, especially about figure 1 and S1 that, for me, could be better described and be coherent with each others :

l.6 : instead of "the maximum depth of the thermocline" what you compute more exactly is "the maximum thermocline depth" or identically "the depth of the maximum thermocline"

Mistake corrected (line 6).

l. 102-103: I find the formulation here a bit unclear. (i) I would suggest instead something like : "To locate the MTD, we computed the vertical maximum of the contribution of temperature to the squared brunt-Vaisala frequency (i.e. maximum of $N_T^2$) to locate the most stratified point from the temperature profile." (ii) Also, it is quite obvious what is $N_T^2$ but still there is no explicit routine to compute it in TEOS10 as far as I have checked, so it could be nice to recall the formulation of $N_T^2$ = - g2 * rho * alpha * d(Cons.Temp.)/d(press).

(i) Suggestion accepted (lines 105-107).

(ii) Suggestion accepted (lines 108-110 and Eq. (1)).

*"$N_T^2$ is given by Eq. (1), where $g$ is the gravitational acceleration, $\rho$ is the density, $\alpha^\Theta$ is the coefficient of thermal expansion, $\Delta\Theta$ is the vertical conservative temperature gradient, and $\Delta P$ is the pressure gradient.*

$$N_T^2 = g^2 \rho \frac{-\alpha^\theta \Delta\theta}{\Delta P}$$  (1)"

l.214-215 : this sentence is not clear grammatically I think, and should be rephrased

We modified this paragraph, hoping to be clearer (lines 188-190).

l.338 : "Tin" à "In"

Mistake corrected (line 292).

l.340 : "thermocline"

Mistake corrected (line 295).

(i) Figure 1 : when comparing figure 1 with figure S1 (which is a plus to have it), I do not find the two corresponding upper/lower thermocline limits sometimes and also the colour code is not the same it seems (upper/lower is resp. black/red in fig1 and in figS1, it looks to be the opposite, so you should arrange this to be same). For exemple, on fig S1b, there is only one black dotted line which seems to be the one around 40m but the 400m one is not there, why ? better to have it in both figures , same for figS1c (missing red line) ; same for fig1g which has only the black line while figS1g has also one red one close to surface ; same for figS1f which has very different lines that on fig1f …. (ii) Also the legend of fig1 and figS1 should describe further what are those lines. You write about thermocline and mld or barrier limits on fig S1 but you never say to what they correspond on the figure. This should be corrected as for now it is unclear and misleading to me.

(i) Figures 1 and S1 did not have the same color code. We have corrected Fig. S1 from the supplementary information to avoid confusion.

(ii) Suggestion accepted. Now, in addition to the box in panel (a) of Fig. 1 and S1, we indicate in the caption what each of the lines and points of the figures are.

Figure 8 : missing unit of the variable above the colorbar, can be e.g. "Thermocline Strength Index (TSI) [degC/m]"

Suggestion accepted (line 162 and Fig. 8).

####################################################################################

---

## Author Response (AR3)

**Reply to Editor**

Response to the editor comments

The Authors thank the editor for their comments that have helped to improve our manuscript. We hope that the editor finds our manuscript now suitable for publication in Ocean Science. Hereinafter, the reviewer's comments are in black and the authors' answers in blue.

Very minor textual changes

- Suggestions and typos were accepted and corrected.

- Units were changed to Roman font.

- The Equations were manually centered. We are using the Copernicus Publications Manuscript Preparation Template for LaTeX Submissions and the "amsthm" package, but they do not automatically center the Equations.

- The $N^2$ equation we are using to describe $N_T^2$ is the one found in the TEOS-10 manuals (https://www.teos-10.org/pubs/TEOS-10_Manual.pdf and https://www.teos-10.org/pubs/gsw/pdf/Nsquared.pdf). We understand that there might be confusion, so we have added the citation and changed the description to $\Delta\theta$ and $\Delta P$, hoping to be more clear.